# Diffusion-based Reinforcement Learning via Q-weighted Variational Policy Optimization

**Shutong Ding**[1,3]   **Ke Hu**[1]   **Zhenhao Zhang**[1]   **Kan Ren**[1,3]   **Weinan Zhang**[2]
**Jingyi Yu**[1,3]   **Jingya Wang**[1,3]   **Ye Shi**[1,3] *

[1]ShanghaiTech University   [2]Shanghai Jiao Tong University
[3]MoE Key Laboratory of Intelligent Perception and Human Machine Collaboration

```
{dingsht, v-huke, v-zhangzhh1, renkan}@shanghaitech.edu.cn
     {yujingyi, wangjingya, shiye}@shanghaitech.edu.cn
                 wnzhang@sjtu.edu.cn
```

## Abstract

Diffusion models have garnered widespread attention in Reinforcement Learning (RL) for their powerful expressiveness and multimodality. It has been verified that utilizing diffusion policies can significantly improve the performance of RL algorithms in continuous control tasks by overcoming the limitations of unimodal policies, such as Gaussian policies. Furthermore, the multimodality of diffusion policies also shows the potential of providing the agent with enhanced exploration capabilities. However, existing works mainly focus on applying diffusion policies in offline RL, while their incorporation into online RL has been less investigated. The diffusion model's training objective, known as the variational lower bound, cannot be applied directly in online RL due to the unavailability of 'good' samples (actions). To harmonize the diffusion model with online RL, we propose a novel model-free diffusion-based online RL algorithm named Q-weighted Variational Policy Optimization (QVPO). Specifically, we introduce the Q-weighted variational loss and its approximate implementation in practice. Notably, this loss is shown to be a tight lower bound of the policy objective. To further enhance the exploration capability of the diffusion policy, we design a special entropy regularization term. Unlike Gaussian policies, the log-likelihood in diffusion policies is inaccessible; thus this entropy term is nontrivial. Moreover, to reduce the large variance of diffusion policies, we also develop an efficient behavior policy through action selection. This can further improve its sample efficiency during online interaction. Consequently, the QVPO algorithm leverages the exploration capabilities and multimodality of diffusion policies, preventing the RL agent from converging to a sub-optimal policy. To verify the effectiveness of QVPO, we conduct comprehensive experiments on MuJoCo continuous control benchmarks. The final results demonstrate that QVPO achieves state-of-the-art performance in terms of both cumulative reward and sample efficiency. Our official implementation is released in `https://github.com/wadx2019/qvpo/`.

## 1 Introduction

Recent years have witnessed significant success in the application of diffusion policy in imitation learning [6, 22, 27, 32], and offline reinforcement learning [40, 1, 16, 4, 5, 11, 44, 26]. In imitation

---

*Corresponding author.

38th Conference on Neural Information Processing Systems (NeurIPS 2024).

learning, diffusion models [13, 35, 7] are often employed to capture the intricate distributions found within expert datasets. Furthermore, in offline RL, certain works replace unimodal policies with diffusion models to enrich the diversity in sampled actions or decision sequences. This success can be largely attributed to the robust expressiveness and multimodality inherent in diffusion models [13, 35]. Such methods have demonstrated their effectiveness in practical scenarios like robotic navigation [36, 43], robot arm manipulation[6, 17, 33, 23], dexterous hand and legged robot locomotion control [14].

While diffusion policies have been extensively explored, their application in online RL has received relatively less attention. In online RL, value estimation varies with policy changes [45], posing a significant challenge. While diffusion models can effectively capture data distribution, they cannot directly improve the policy due to their training objectives [42]. Consequently, integrating diffusion models into traditional online RL framework [34, 10, 9, 38] is challenging. Some works, such as DIPO [42], propose leveraging Q-function gradients to update actions for higher rewards, followed by employing diffusion models to fit the updated action distribution. However, relying on gradient updates constrains the algorithm's exploration capability. QSM [31] directly aligns the score and the gradient of the Q-function under the perspective of score-matching. However, since the gradient of the Q-network is inaccurate, QSM has a doubled approximation error from the alignment process, which prevents the policy from converging to optimality.

To address these issues, we propose a novel model-free online RL algorithm called Q-weighted Variational Policy Optimization (QVPO). The core idea behind QVPO is straightforward yet effective. By revisiting the VLO objective of diffusion models and the policy objective of online RL, we discovered that the Variation LOwer Bound (VLO) objective, with appropriate weights for state-action pairs, can form a tight lower bound of the policy objective under certain conditions. This new objective, termed Q-weighted VLO loss, is complemented by Q-weight transformation functions, allowing it to be applied to general RL tasks. Additionally, we add an entropy term to the diffusion policy loss to enhance exploration. Calculating the exact entropy is intractable due to the inaccessibility of the probability density function (PDF) of diffusion policies. Therefore, we design an entropy regularization term from another perspective. Moreover, while diffusion policies offer advantages like multimodality, they also introduce large policy variance, leading to inefficient interaction with environments. To address this, we develop an efficient behavior policy through action selection to improve sample efficiency. With QVPO, diffusion policies can fully leverage their exploration capability and multimodality in the online RL paradigm. QVPO achieves state-of-the-art performance in terms of both cumulative reward and sample efficiency compared to traditional online RL methods and existing diffusion-based RL methods in MuJoCo locomotion tasks [39].

Our contributions are summarized as follows:

1) **Q-weighted VLO Loss.** By revisiting the VLO loss of diffusion models and policy loss in online RL, we propose the Q-weighted VLO loss, which is the core component of the proposed QVPO method. We further extend its application to general RL tasks via equivalent Q-weight transformation functions. Additionally, we theoretically prove that the Q-weighted VLO loss is a tight lower bound of the policy objective in online RL.

2) **Diffusion Entropy Regularization & Efficient Behavior Policy.** We find that the exploration ability of diffusion policies declines with limited diffusion steps. To address this, we apply a special entropy regularization term to the policy loss, which is nontrivial for diffusion policies. Additionally, to cope with the large policy variance of the diffusion model, we develop an efficient behavior policy through action selection to improve sample efficiency.

3) **State-of-the-art Performance.** We verified the effectiveness of QVPO on MuJoCo locomotion benchmarks. Experimental results indicate that QVPO achieves state-of-the-art performance in terms of sample efficiency and episodic reward compared to both previous traditional and diffusion-based online RL algorithms.

## 2   Related Works

In this section, we will review the existing works that utilize diffusion models in decision-making tasks and generally divide them into four categories according to different applications.

**Diffusion Policies for Imitation Learning.** Imitation learning is a framework for learning a behavior policy from expert datasets without explicit reward. Diffusion models can effectively fit the distribution of given datasets due to their powerful expressiveness. Some research works such as Diffusion

Policy [6], Crossway Diffusion [22], AVDC [18] and [27] exemplify this by generating robot action sequences via diffusion based policy conditioned on visuomotor observation. Considering the reward is sparse or inaccessible in the expert dataset, BESO [32] leverages the diffusion model in the domain of goal-conditioned imitation learning to learn a goal-specified policy.

**Diffusion Planners.** Planning in RL involves using the dynamic function to inform decision-making over a long horizon. Diffusion planners reduce compounding errors by directly constructing the complete trajectory rather than relying on one-step transitions. Diffuser [15] arranges state-action sequences into a structured two-dimensional array and samples the trajectories based on reward guidance. In subsequent work, Decision Diffuser [2] generates state sequences with classifier-free guidance and then uses inverse dynamic function to derive actions to alleviate the distribution shift problem. Latent Diffuser [20] generates sequences in a learned latent space and then reconstructs the original trajectories. A bunch of works such as [21, 3, 8] leverage hierarchical structures to enhance learning efficiency and decision-making capabilities by decomposing complex tasks into a hierarchy of sub-tasks.

**Diffusion Policies for Offline RL.** Unlike most imitation learning problems without reward and assuming an optimal expert to provide the data, offline RL faces the challenge of learning optimal policy from suboptimal offline datasets [19]. Diffusion-QL [40] integrates the diffusion model with the conventional Q-learning framework. However, the training of Diffusion-QL is unstable in the out-of-distribution (OOD) state region, SRDP [1] reconstructs the state to alleviate the distribution shift caused by OOD. EDP [16] reduces the computation cost during diffusion inference by applying Tweedie's formula to predict actions. CEP [24] draws samples by diffusion sampling with exact guidance defined by the energy function based on contrastive learning. CPQL [5] uses a consistency model to represent the policy. Some works [12, 4, 11] utilize diffusion models to construct the optimal policy by weighted regression [29, 28]

**Diffusion Policies for Online RL.** Online RL faces a more challenging problem in that it requires algorithms to learn the policy and make decisions in real-time interaction with the environment. DIPO [42] is the first to employ diffusion policies in online RL and proposes a novel policy improvement method. During diffusion policy updates, each action in the reply buffer is updated via action gradient to receive higher rewards. However, the action of gradient updating may deviate from the behavior policy, and the paradigm of gradient updating limits the algorithm's ability of global exploration and raises computational costs. QSM [31] introduces a new update role for diffusion policy by aligning the score of the diffusion model with the gradient of the Q-function to increase the Q-value of state-action pairs. However, it overlooks the impact of inaccuracies in the value function gradient and biases introduced during the alignment process when updating the policy. This can lead to the policy being influenced by suboptimal data within the buffer. Compared with these existing diffusion-based online RL algorithms, QVPO performs policy optimization with the Q-weighted VLO loss which can be theoretically proven to be the tight lower bound of the policy objective of online RL. This means QVPO does not incur additional errors in policy optimization. Besides, diffusion policy entropy regularization and efficient behavior policy via action selection techniques are provided to further enhance the performance of the diffusion policy in online RL.

## 3 Preliminaries

### 3.1 Reinforcement Learning

For reinforcement learning (RL), an MDP is defined as $(\mathcal{S}, \mathcal{A}, p, r, \rho_0, \gamma)$, where $\mathcal{S}$ is the state space, $\mathcal{A}$ is the action space, $p : \mathcal{S} \times \mathcal{S} \times \mathcal{A} \to [0, \infty)$ is the transition probability function of the next state $s_{t+1}$ given the current state $s_t$ and the action $a_t$, $r : \mathcal{S} \times \mathcal{A} \to [r_{\min}, r_{\max}]$ is the bounded reward function, $\rho_0 : \mathcal{S} \to [0, \infty)$ is the distribution of the initial state $s_0$ and $\gamma \in [0, 1]$ is the discount factor for value estimation. In an MDP, the process starts from an initial state $s_0$ sampled from $\rho_0$ and then samples actions $a_t$ from the policy $\pi(a|s) : \mathcal{S} \times \mathcal{A} \to [0, \infty)$ given the state $s_t$. Here $\tau = (s_0, a_0, s_1, a_1 \dots)$ denotes this kind of trajectory and $\tau \sim \pi$ denotes the distribution of trajectory $\tau$ given the policy $\pi(a|s)$. The state-value function $V_\pi(s) = \mathbb{E}_{\tau \sim \pi}[\sum_{t=0} \gamma^t r_t | s_0 = s]$ represents the expected reward that the agent can obtain under the policy $\pi$ in the state $s$. Compared to the state-value function, the action-state function is $Q_\pi(s, a) = \mathbb{E}_{\tau \sim \pi}[\sum_{t=0} \gamma^t r_t | a_0 = a, s_0 = s]$ where action $a_0$ is provided as input. Further, the advantage function refers to the extra benefit of taking a specific action relative to taking the average action at a given state which is defined as the following: $A_\pi(s, a) = Q_\pi(s, a) - V_\pi(s)$. The goal of RL is to learn the policy that maximizes the discounted expected

cumulative reward, defined as $J(\pi) = \mathbb{E}_{\tau \sim \pi}[\sum_{t=0} \gamma^t r_t]$, which can be optimized via performing multiple steps of policy improvement $\pi_{k+1} = \arg\max_\pi \mathbb{E}_{\tau \sim \pi_k}[Q_{\pi_k}(s, a) \log(\pi(a \mid s))]$.

## 3.2 Denoising Diffusion Probabilistic Models

Denoising diffusion probabilistic models (DDPM) [13] are powerful latent variable generative models that transform any data distribution into a simple Gaussian distribution by adding noise (forward process) and then denoise it using neural networks (reverse process). Given a dataset $\{\mathbf{x}_0^i\}_{i=1}^N$ for $\mathbf{x}_0^i \sim q(\mathbf{x}_0)$, the forward process of DDPM transforms the distribution $q(x_0)$ into a tractable prior Gaussian distribution by incorporating Gaussian noise in $T$ steps with the transitions:

$$q(\mathbf{x}_t \mid \mathbf{x}_{t-1}) := \mathcal{N}(\mathbf{x}_t; \sqrt{1 - \beta_t}\mathbf{x}_{t-1}, \beta_t \mathbf{I}), \tag{1}$$

where $\beta_t$ is the variance schedule. Using the Markov chain property, we can obtain the distribution of $x_t$ conditioned on $x_0$:

$$q(\mathbf{x}_t \mid \mathbf{x}_0) = \mathcal{N}(\mathbf{x}_t; \sqrt{\bar{\alpha}_t}\mathbf{x}_0, (1 - \bar{\alpha}_t)\mathbf{I}), \tag{2}$$

where $\alpha_t = 1 - \beta_t$ and $\bar{\alpha}_t = \prod_{s=0}^t \alpha_s$. Eventually, when $T \to \infty$, $x_T$ converges to an isotropic Gaussian distribution. In the reverse process, DDPM first generates noise from the prior distribution $p(\mathbf{x}_T) = \mathcal{N}(\mathbf{x}_T; 0, \mathbf{I})$ and then gradually denoises it by learning parameterized transitions $p_\theta(\mathbf{x}_{t-1} \mid \mathbf{x}_t) = \mathcal{N}(\mathbf{x}_{t-1}; \mu_\theta(\mathbf{x}_t, t), \Sigma_\theta(\mathbf{x}_t, t))$ to fit $q(\mathbf{x}_t \mid \mathbf{x}_{t-1}, \mathbf{x}_0)$, where $\theta$ denotes the learnable parameters. The training objective of DDPM is to maximize the Variational LOwer Bound (VLO) defined as $L_{\text{VLO}} = \mathbb{E}_{q(\mathbf{x}_{0:T})} \left[\log \frac{p_\theta(\mathbf{x}_{0:T})}{q(\mathbf{x}_{1:T}|\mathbf{x}_0)}\right]$. Finally, with $\epsilon \sim \mathcal{N}(0, \mathbf{I})$, the loss in DDPM takes the form of:

$$\mathbb{E}_{t \sim [1,T], \mathbf{x}_0, \boldsymbol{\epsilon}_t} \left[||\boldsymbol{\epsilon}_t - \boldsymbol{\epsilon}_\theta(\sqrt{\bar{\alpha}_t}\mathbf{x}_0 + \sqrt{1 - \bar{\alpha}_t}\boldsymbol{\epsilon}_t, t)||^2\right]. \tag{3}$$

# 4 Q-weighted Variational Policy Optimization

In this section, we will first revisit the VLO loss [13] of the diffusion model and the policy objective of online RL [37]. Then, we derive the Q-weighted VLO loss, which is the principal component of the proposed Q-weighted Variational Policy Optimization (QVPO). However, the Q-weighted VLO loss cannot be directly applied in general RL tasks. To address this issue, we also provide equivalent Q-weight transformation functions. Since the derived Q-weighted VLO loss is the tight lower bound of the policy loss in online RL, QVPO can effectively avoid the drawbacks of existing online RL methods for diffusion policies [42, 31] mentioned in Section 2.

QVPO also involves practical techniques for the underlying issues of optimizing diffusion policies. Firstly, to further enhance the exploration ability of diffusion policy, we develop a special entropy regularization. This is not trivial for diffusion policy since the log probability of the state-action pair is not available. Besides, we devise an efficient behavior policy via action selection for the diffusion model to avoid the large policy variance of diffusion, which results in sample inefficiency. With these practical techniques, diffusion policy can achieve better performance with fewer online interactions with the environments. Finally, the training procedure of QVPO is shown in Figure 1.

## 4.1 Q-weighted Variational Objective for Diffusion Policy

As mentioned in DIPO [42], optimizing diffusion policies in the online RL paradigm is nontrivial, which principally results from two reasons. In one aspect, if we directly apply the deterministic policy gradient to diffusion policies like Diffusion-QL [40], the backpropagation chain through the denoising procedure of the diffusion model becomes quite long. This leads to high computational costs and instability during training, which severely restricts the performance of diffusion policies in online RL. In another aspect, samples from the optimal policy are required when directly using the variational bound objective to train diffusion policies. However, these optimal samples are typically unavailable in online RL.

In this context, we revisited the VLO objective of diffusion models and the RL policy objective. To our surprise, we discovered that by adding the appropriate weights to the VLO objective, it becomes a tight lower bound of the RL policy objective under certain conditions.

**Theorem 1.** *(**Lower Bound of RL Policy Objective**) If $Q(s, a) \geq 0$ for any state-action pair $(s, a)$, the Q-weighted variational bound objective of diffusion policy*

$$\mathbb{E}_{s, a \sim p(s'|s,a), \pi_k(a|s)} \left[Q(s, a_0) \cdot \mathbb{E}_{a_{1:T} \sim q(a_{1:T}|s, a_0)} \left[\log \frac{\pi_\theta(a_{0:T} \mid s)}{q(a_{1:T} \mid s, a_0)}\right]\right]$$

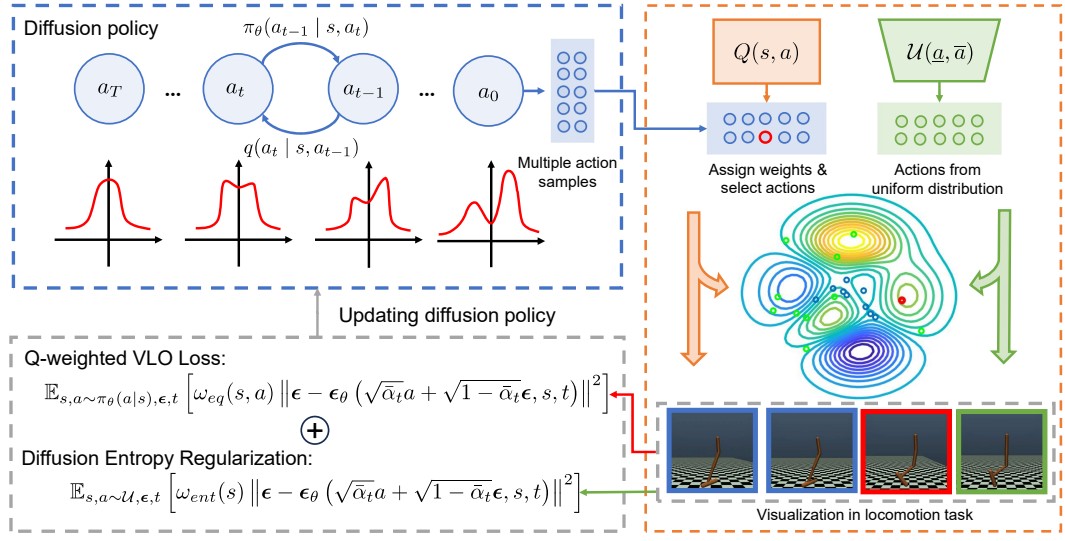

Figure 1: The training pipeline of QVPO. In each training epoch, QVPO first utilizes the diffusion policy to generate multiple action samples for every state. Then, these action samples will be selected and endowed with different weights according to the Q value given by the value network. Besides, action samples from uniform distribution are also created for the diffusion entropy regularization term. With these action samples and weights, we can finally optimize the diffusion policy via the combined objective of Q-weighted VLO loss and diffusion entropy regularization term.

*is the tight lower bound of the objective of RL policy*

$$\mathbb{E}_{s,a\sim p(s'|s,a),\pi_k(a|s)}\left[Q(s,a)\log(\pi_\theta(a|s))\right],$$

*and the equality holds when the policy converges.*

The proof can be referred to in the supplementary materials. According to this theorem, we can derive the Q-weighted variational bound loss as (4), which can be applied to diffusion policy optimization.

$$\mathcal{L}(\theta) \triangleq \mathbb{E}_{s,a\sim\pi_k(a|s),\boldsymbol{\epsilon},t}\left[\frac{\beta_t^2}{2\sigma_t^2\alpha_t\left(1-\bar{\alpha}_t\right)}Q(s,a)\cdot\left\|\boldsymbol{\epsilon}-\boldsymbol{\epsilon}_\theta\left(\sqrt{\bar{\alpha}_t}a+\sqrt{1-\bar{\alpha}_t}\boldsymbol{\epsilon},s,t\right)\right\|^2\right]. \quad (4)$$

According to [13], removing the coefficient $\frac{\beta_t^2}{2\sigma_t^2\alpha_t(1-\bar{\alpha}_t)}$ does not affect the training of diffusion. Hence, the final Q-weighted VLO loss is defined as

$$\mathcal{L}(\theta) \triangleq \mathbb{E}_{s,a\sim\pi_k(a|s),\boldsymbol{\epsilon},t}\left[Q(s,a)\cdot\left\|\boldsymbol{\epsilon}-\boldsymbol{\epsilon}_\theta\left(\sqrt{\bar{\alpha}_t}a+\sqrt{1-\bar{\alpha}_t}\boldsymbol{\epsilon},s,t\right)\right\|^2\right]. \quad (5)$$

While we can now use (5) to optimize the diffusion policy, two issues remain to be resolved.

1) **Negative $Q$ value.** In real-world decision-making tasks, it is difficult to ensure that the returned reward is always non-negative, which means the $Q$ value may be negative for some state-action pairs. Therefore, to apply QVPO in practical tasks, we must address situations where the $Q(s,a)$ value is negative for certain states and actions.

2) **High-quality Training Samples.** According to (5), achieving significant policy improvement requires obtaining certain rare state-action samples with high Q values. However, this is a significant challenge with limited interaction with the environment. This issue is also common in other online RL methods based on weighted policy training, such as RWR [30], and it hinders their application to real-world tasks.

### 4.2 Equivalent Transformation for Q-weighted VLO Loss

To resolve these two problems in training diffusion policies with the Q-weighted VLO loss, we propose equivalent Q-weight transformation functions to convert the Q value into an equivalent positive Q weight and leverage the powerful data-synthesizing ability of the diffusion model to generate high-quality samples for training. The new equivalent Q-weighted VLO loss is defined in (6). We will introduce the detailed implementation of these solutions in the following contexts.

$$\mathcal{L}(\theta) \triangleq \mathbb{E}_{s,a\sim\pi_k(a|s),\boldsymbol{\epsilon},t}\left[\omega_{eq}(s,a)\left\|\boldsymbol{\epsilon}-\boldsymbol{\epsilon}_\theta\left(\sqrt{\bar{\alpha}_t}a+\sqrt{1-\bar{\alpha}_t}\boldsymbol{\epsilon},s,t\right)\right\|^2\right]. \quad (6)$$

**Theorem 2.** *(Optimal Solution in One Policy Improvement Step) Considering the optimization problem in policy improvement step* $\pi_{k+1} = \arg\max_\pi \mathbb{E}_{\tau \sim \pi_k}[Q_{\pi_k}(s, a) \log(\pi(a \mid s))]$, *the optimal policy in a given state* $s$ *is*

$$p(a \mid s) = \begin{cases} \frac{\pi(a|s)Q(s,a)}{\int \mathbb{I}_{Q(s,a)>0}(a)\pi(a|s)Q(s,a)da}, & Q(s,a) > 0 \\ 0, & \text{otherwise} \end{cases} \tag{7}$$

*when there exists* $a$ *such that* $Q(s,a) > 0$, *and*

$$p(a \mid s) = \frac{1}{N} \sum_{i=1}^{N} \delta(x - a_i), a_i \in \left\{ a \mid Q(s,a) = \max_a Q(s,a), \pi(a \mid s) > 0 \right\}, \tag{8}$$

*when* $Q(s,a) \leq 0$ *for any* $a \in \mathcal{A}$, $\delta(x - a)$ *is the Dirac function,* $N$ *is the cardinality of set* $\{a \mid Q(s,a) = \max_a Q(s,a), \pi(a \mid s) > 0\}$, $\pi(a \mid s)$ *is the behavior policy,* $p(a \mid s)$ *is the policy to be optimized and* $\mathbb{I}_{Q(s,a)>0}$ *is the indicator function that judges whether* $Q(s,a) > 0$.

The proof can be referred to in the supplementary materials. Hence, with Theorem 2, we can obtain the probability density function (PDF) of the optimal policy in one policy improvement step and further use $\frac{p(a|s)}{\pi(a|s)}$ as the weight to achieve an equivalent optimized diffusion policy according to (6). However, the difficulty of judging whether any $Q(s,a) \leq 0$ in a certain state $s$ is still a problem. Therefore, we first come up with an approximate solution "*qcut*" weight transformation function, which returns $Q(s,a)$ when $Q(s,a) \geq 0$ and a small value $\epsilon > 0$ when $Q(s,a) < 0$. Furthermore, the "*qcut*" weight transformation function just endows the best action with the weight mentioned above and discards the left action samples via setting them as zero.

In short, *qcut* weight transformation function takes the formulation of the above two situations into account and "cuts" them via a small value $\epsilon > 0$. Notably, we remove the denominator $\int \mathbb{I}_{Q(s,a)>0}\pi(a \mid s)Q(s,a)da$ in *qcut* weight transformation function. This is because the denominator $\int \mathbb{I}_{Q(s,a)>0}(a)\pi(a \mid s)Q(s,a)da$ is somewhat like the $V(s)$, which denotes the "importance" of the state $s$. Therefore, removing this item can allow important states to obtain larger weights during training, which is confirmed in practice.

However, we find that the *qcut* weight transformation function does not work very well when Q values in most state-action pairs are less than zero. To address this issue, we propose another "*qadv*" weight transformation function, which is much simpler but performs better in our experiments.

$$\omega_{eq}(s,a) \triangleq \omega_{qadv}(s,a) = \begin{cases} A(s,a), & A(s,a) \geq 0 \\ 0, & A(s,a) < 0 \end{cases} \tag{9}$$

where $A(s,a) = Q(s,a) - V(s)$ is the advantage function. *qadv* weight transformation function replaces the $Q(s,a)$ with the advantage $A(s,a)$ to avoid the second situation. Moreover, applying Q value and advantage in policy optimization has been proved equivalent and can also reduce the training variance in previous works like [25].

It is worth noting that, for *qadv* weight transformation function, we do not require an extra network to approximate $V(s)$ to calculate $A(s,a)$. We can always sample a certain number of actions for a state $s$ with diffusion policy and estimate $V(s)$ using the average of their $Q(s,a)$. Besides, for *qadv* function, we also recommend just selecting the best action sample with the largest A value for each state in policy optimization, which can further improve the quality of training samples and reduce the training cost.

**Remark.** Although the weighted VLO loss of QVPO looks similar to AWR [28] and EDP [16], they are different intrinsically. With the Q-weighted VLO loss, QVPO trains the diffusion policy with the weighted sample from the current policy, while AWR or EDP trains the diffusion policy with the weighted sample from the policy in the offline dataset (replay buffer). These "*on-policy*" samples truly benefit the online training. Moreover, AWR and EDP utilize the $\exp$ weight function; but this function is too conservative for online RL.

### 4.3 Enhancing Policy Exploration via Diffusion Entropy Regularization

While diffusion policies can achieve impressive performance with the Q-weighted VLO loss in online RL, their powerful exploration capabilities have yet to be fully harnessed. As noted in Diffusion-QL

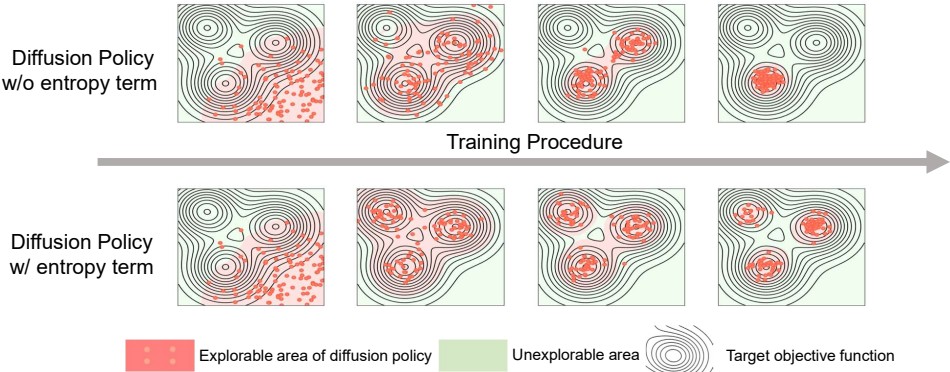

Figure 2: A toy example on continuous bandit to show the effect of diffusion entropy regularization term via the changes of the explorable area for diffusion policy with the training procedure. The contour lines indicate the reward function of continuous bandit, which is an arbitrarily selected function with 3 peaks.

[40], the policy expressiveness of the diffusion model decreases with fewer diffusion steps. In our experiments, we find that not only does policy expressiveness decline, but the exploration capability of the diffusion model also diminishes when the number of diffusion steps is reduced. However, it is essential to limit the number of diffusion steps to avoid excessive training and evaluation costs in real-world applications. Therefore, studying how to enhance the exploration ability of the diffusion model with a limited number of diffusion steps is necessary.

Adding an extra entropy regularization term to the policy loss appears to be a good solution, as it has been validated for categorical policies in discrete action spaces and Gaussian policies in continuous action spaces. However, estimating the entropy for a diffusion policy is nontrivial due to the inaccessibility of the log-likelihood of action samples. Moreover, maximizing the entropy of a policy can be viewed as narrowing the distance between the policy and a maximum entropy distribution (i.e., uniform distribution) in some sense. Thus, by recalling the VLO loss of the diffusion model, the entropy of a diffusion model can be increased with training samples from the uniform distribution. Based on this idea, we propose an entropy regularization term for diffusion policies as follows,

$$\mathcal{L}_{ent}(\theta) \triangleq \mathbb{E}_{s,a\sim\mathcal{U},\boldsymbol{\epsilon},t}\left[\omega_{ent}(s)\left\|\boldsymbol{\epsilon}-\boldsymbol{\epsilon}_\theta\left(\sqrt{\bar{\alpha}_t}a+\sqrt{1-\bar{\alpha}_t}\boldsymbol{\epsilon},s,t\right)\right\|^2\right], \tag{10}$$

where $\omega_{ent}(s) = \omega_{ent}\sum_{i=1}^{N}\frac{\omega_{eq}(s,a_i)}{N}$ is the coefficient related to state $s$ for balancing the trade-off between exploitation and exploration. Here, $N$ represents the number of selected training samples from the diffusion policy for state $s$. When $\omega_{ent}$ is large, the exploration ability of the diffusion policy is improved, whereas the exploitation ability is reduced. Figure 2 is the experiment results on a continuous bandit toy example that clearly illustrates the effect of this entropy regularization term on the diffusion policy, and the concrete reward function of this bandit problem is $R(x) = \sum_{i=1}^{3}w_i\frac{1}{2\pi\sigma_i^2}\exp\left(-\frac{1}{2\sigma_i}(x-\mu_i)^T(x-\mu_i)\right)$, where $w_i = 1.5$, $\sigma_i = 0.1$, and $\mu_i = [-1.35, 0.65]^T$, $[-0.65, 1.35]^T$ and $[-1.61, 1.61]^T$ respectively.

## 4.4 Reducing Diffusion Policy Variance via Action Selection

Despite the diffusion model that allows the online RL agent to seek better policies, it also introduces a large policy variance. This results in inefficiency for online interactions of the behavior policy with the environment. To address this issue, we propose an efficient behavior policy via action selection for diffusion model $\pi_\theta^K(a \mid s)$, which improves the sample efficiency. Specifically, it is motivated by the idea that the efficiency of action samples from behavior policy depends on their Q values. In other words, action samples from behavior policy with high Q values indicate that the following trajectory also has a large underlying reward and is of great significance for training. Specifically, the efficient behavior policy $\pi_\theta^K(a \mid s)$ is defined as:

$$\pi_\theta^K(a \mid s) \triangleq \underset{a\in\{a_1,\cdots,a_K\sim\pi_\theta(a|s)\}}{\text{argmax}} Q(s,a).$$

Furthermore, while using the efficient behavior policy $\pi_\theta^K(a \mid s)$ as a behavior policy can improve sample efficiency, it is not recommended to apply the same action selection number $K$ to the target

policy for calculating the TD target. This is due to the large policy variance of the diffusion policy, which can result in a serious overestimation of the target Q value. Hence, if we choose the action selection number $K_b$ for the behavior policy, a smaller action selection number $K_t < K_b$ for the target policy is often a good choice in practice.

Table 1: Comparison of QVPO and 6 other online RL algorithms in evaluation results. (N/A indicates the algorithm does not work)

| Envrironments | PPO | SPO | TD3 | SAC | DIPO | QSM | QVPO(*) |
|---|---|---|---|---|---|---|---|
| Hopper-v3 | 3154.3(426.2) | 2212.8(988.4) | 3267.5(8.5) | 2996.6(111.9) | 3295.4(7.0) | 2154.7(998.2) | **3728.5(13.8)** |
| Walker2d-v3 | 3751.5(609.1) | 3321.8(1328.9) | 3513.9(40.7) | 4888.1(80.0) | 4681.7(25.7) | 3613.4(1443.5) | **5191.8(60.2)** |
| Ant-v3 | 2781.9(74.1) | 2100.2(302.4) | 4583.8(69.5) | 5030.9(1000.3) | 5665.9(54.7) | N/A | **6425.1(67.6)** |
| HalfCheetah-v3 | 4773.5(53.4) | 4008.2(246.8) | 10388.6(80.4) | 10616.9(72.8) | 9590.5(67.5) | 3888.2(632.6) | **11385.6(164.5)** |
| Humanoid-v3 | 713.7(85.9) | 797.4(262.1) | **5353.5(53.7)** | 5159.7(475.3) | 4945.5(898.6) | 4793.1(229.5) | 5306.6(14.5) |

## 5  Experiments

In this section, we first describe a practical implementation of QVPO, as shown in Algorithm 1, which is implemented in an off-policy fashion. Then, we evaluate the proposed algorithm on different decision-making tasks and with various hyper-parameters. With these experimental results, we aim to answer three questions:

- How does QVPO compare to previous popular online RL algorithms and existing diffusion-based online RL algorithms?
- How does the entropy regularization term affect the performance of QVPO?
- Can the $K$-efficient behavior policy significantly improve the sample efficiency of QVPO?

---

**Algorithm 1** Q-weighted Variational Policy Optimization

---

**Input:** Diffusion policy $\pi_\theta(a \mid s)$, value network $Q_\omega(s, a)$, replay buffer $\mathcal{D}$, $K_b$-effiecient diffusion policy for behavior policy, $K_t$-effiecient diffusion policy for target policy, number of training samples $N_d$ from diffusion policy, number of training samples $N_e$ from uniform distribution $\mathcal{U}(\underline{a}, \overline{a})$.

1: **for** $t$ **in** $1, 2, \cdots, T$ **do**
2:     Sample the action using the diffusion policy $\pi_\theta^{K_b}(a \mid s_t)$.
3:     Take the action $a_t$ in the environment and store the returned transition in $\mathcal{D}$.
4:     Sample a mini-batch $\mathcal{B}$ of transitions in $\mathcal{D}$.
5:     Generate $N_d$ samples from $\pi_\theta(a \mid s)$, and $N_e$ samples from $\mathcal{U}(\underline{a}, \overline{a})$ for each state $s$ in $\mathcal{B}$.
6:     Endow the $N_d$ samples with weights (9).
7:     Select an action sample $a_{max}$ with maximum weight among $N_d$ samples for training.
8:     Endow the $N_e$ samples with the weight $\omega_{ent}(s) = \omega_{ent} \cdot \omega_{eq}(s, a_{max})$.
9:     Update the parameters of the diffusion policy using the summation of (6) and (10).
10:     Construct TD target as $y_t = r_t + \gamma Q_\omega(s_{t+1}, \pi_\theta^{K_t}(a \mid s_{t+1}))$ for each $(s_t, a_t, r_t, s_{t+1})$ in $\mathcal{B}$.
11:     Update the parameters of the value network using MSE loss.
12: **end for**

---

### 5.1  Comparative Evaluation

To demonstrate the effectiveness of our method, we compared QVPO with six other online model-free RL algorithms: two off-policy algorithms (TD3 [9] and SAC [10]), two on-policy algorithms (PPO [34] and SPO [3]), and one advanced diffusion-based online RL algorithm (DIPO [42], QSM [31]). These comparisons were conducted on five MuJoCo locomotion tasks [39]. we plotted the learning curves of QVPO and the six other algorithms over five random runs, as shown in Figure 3. The solid curve represents the average return, and the transparent shaded region represents the standard deviation. Each experiment was conducted over $1e6$ training epochs. According to the learning curves, QVPO achieves state-of-the-art performance compared to the other six algorithms, and converges much faster than the other algorithms, further demonstrating its sample efficiency. Additionally, the practical implementation of QVPO follows SAC in critic part, which utilizes doubled Q networks and only use the minimum for policy and critic update.

Table 1 presents the evaluation results of QVPO and six other algorithms on the MuJoCo locomotion tasks after $1e6$ iterations, further confirming the superiority of QVPO. Notably, during the evaluation

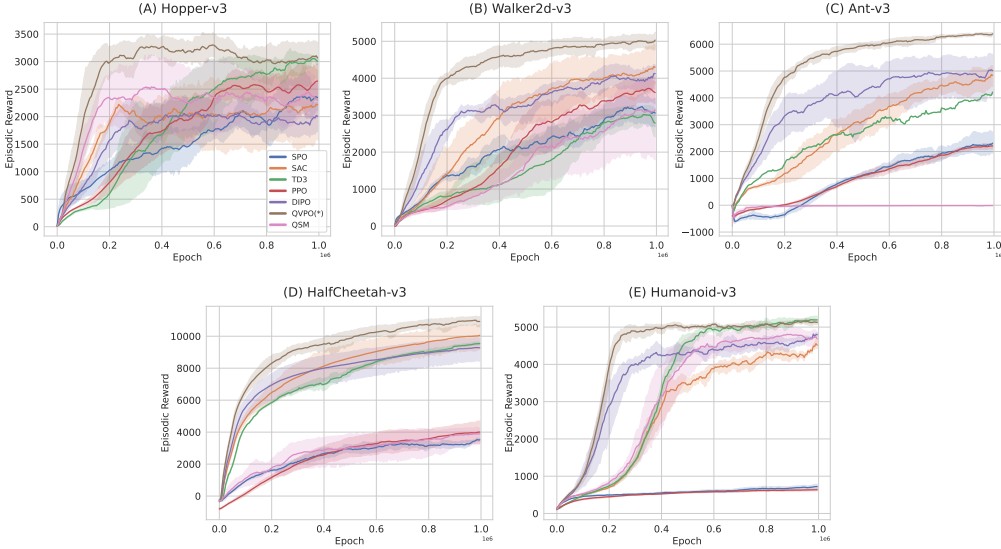

Figure 3: Learning Curves of different algorithms on 5 Mujoco locomotion benchmarks across 5 runs. The x-axis is the number of training epochs. The y-axis is the episodic reward. the plots smoothed with a window of 5000.

stage, we apply the efficient behavior policy with a large action selection number $K = 32$ instead of the deterministic denoising procedure used in DIPO [42]. As illustrated in [41], the high-probability region in each diffusion step is not the origin but a Gaussian sphere. Hence, using the deterministic denoising procedure in the evaluation stage is not a good choice. In that case, using the proposed efficient behavior policy during evaluation is more reasonable. More details related to the experiments can be found in the supplementary materials.

## 5.2 Ablation Study and Parameter Analysis

To further examine the significance of each component in QVPO, we also conducted the ablation study and parameter analysis on the effects of the entropy regularization and the $K$-efficient behavior policy. Here We use the experiment on Ant-v3 as an example since the final results are similar across different locomotion tasks.

**Effect of Diffusion Entropy Regularization.** As shown in Figure 4, without the diffusion entropy regularization loss, QVPO with 100 diffusion steps achieves much better performance than QVPO with 20 diffusion steps. However, when the entropy regularization loss is applied, QVPO with 20 diffusion steps achieves close performance to QVPO with 100 diffusion steps, which verifies the effectiveness of the entropy regularization term.

**Action Selection number $K$ for Efficient Behavior Policy.** Figure 5 presents the performance of QVPO with different action selection numbers for behavior policy $K_b$ and for target policy $K_t$. It can be observed that QVPO with action selection numbers $K_b = 4, K_t = 1$ is superior to QVPO with action selection numbers $K_b = 1, K_t = 1$, while the training stability of QVPO with $K_b = 4, K_t = 2$ and $K_b = 4, K_t = 1$ is better than that of QVPO with $K_b = 4, K_t = 4$. This indicates that efficient behavior policy via action selection can improve sample efficiency and setting the action selection numbers $K_t < K_b$ can help reduce the overestimation error of the target Q value. Besides, the experiment with $K_b = 20, K_t = 2$ shows that a

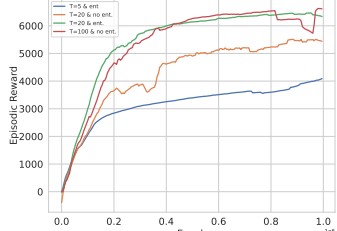

Figure 4: Comparison between QVPO with and without the diffusion entropy regularization.

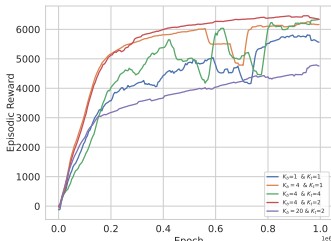

Figure 5: Comparison of QVPO with different action selection numbers for behavior policy $K_b$ and for target policy $K_t$.

too-high action selection number will lead to a limited exploration. Therefore, setting $K_b = 4$ can balance exploration and exploitation well.

## 6 Conclusion

In this paper, we proposed a novel diffusion-based online RL algorithm called QVPO that sufficiently exploits the expressiveness and multimodality of diffusion policy. QVPO leverages the Q-weighted VLO loss, a core component that serves as a tight lower bound of the policy objective in online RL under certain conditions, facilitated by Q-weight transformation functions. Additionally, we designed a special entropy regularization term to enhance the exploration capabilities of diffusion policies and the efficient behavior policy to improve sample efficiency by reducing the behavior policy variance. Our comprehensive experiments on MuJoCo continuous control benchmarks demonstrate that QVPO achieves state-of-the-art performance in terms of both cumulative reward and sample efficiency, surpassing both traditional and existing diffusion-based online RL methods.

However, there are still several underlying challenges in applying diffusion policies to online RL that require further exploration. For instance, although we developed a special entropy regularization loss to approximate the effect of maximizing entropy, it lacks the ability to adaptively adjust the entropy term and cannot incorporate the entropy of the diffusion policy into the TD target for soft policy iteration, as seen in SAC. Future work will focus on developing adaptive entropy adjustment mechanisms and integrating entropy into the TD target to enable soft policy iteration, which we believe will further enhance the performance of diffusion policies in online RL.

## Acknowledgement

This work was supported by NSFC (No.62303319), Shanghai Sailing Program (22YF1428800), Shanghai Local College Capacity Building Program (23010503100), ShanghaiTech AI4S Initiative SHTAI4S202404, Shanghai Frontiers Science Center of Human-centered Artificial Intelligence (ShangHAI), HPC Platform of ShanghaiTech University, MoE Key Laboratory of Intelligent Perception and Human-Machine Collaboration (ShanghaiTech University) and Shanghai Engineering Research Center of Intelligent Vision and Imaging. We also thank Ming Zhou, Shanghai AI Laboratory, for his discussion and suggestion in our proof of Theorem 4.1.

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

# A Proofs

## A.1 Proof of Theorem 1

**Theorem 1.** (***Lower Bound of RL Policy Objective***) *If $Q(s, a) \geq 0$ for any state-action pair $(s, a)$, the Q-weighted variational bound objective of diffusion policy*

$$\mathbb{E}_{s, a \sim p(s'|s,a), \pi_k(a|s)} \left[ Q(s, a_0) \cdot \mathbb{E}_{a_{1:T} \sim q(a_{1:T}|s, a_0)} \left[ \log \frac{\pi_\theta (a_{0:T} \mid s)}{q (a_{1:T} \mid s, a_0)} \right] \right]$$

*is the tight lower bound of the objective of RL policy*

$$\mathbb{E}_{s, a \sim p(s'|s,a), \pi_k(a|s)} \left[ Q(s, a) \log(\pi_\theta(a|s)) \right],$$

*and equality holds when the policy converges.*

*Proof.* The policy objective of online RL is

$$\max_\theta \mathbb{E}_{s, a \sim p(s'|s,a), \pi_k(a|s)} \left[ Q(s, a) \log(\pi_\theta(a|s)) \right],$$

where $\pi_k(a \mid s)$ indicates the behavior policy in state $s$, $\pi_\theta(a \mid s)$ is the policy to be optimized, and $Q(s, a)$ represents the Q value of the state-action pair $(s, a)$. Then, if $\pi_\theta(a \mid s)$ is a diffusion policy, the inner-term $Q(s, a) \log (\pi_\theta(a|s))$ equals to

$$Q(s, a_0) \mathbb{E}_{a_{1:T} \sim q(a_{1:T}|s, a_0)} \left[ \log \frac{\pi_\theta (a_{0:T} \mid s)}{q (a_{1:T} \mid s, a_0)} \right] + Q(s, a_0) \, \mathrm{D}_{KL} \left( q (a_{1:T} \mid s, a_0) \, \| \, \pi_\theta (a_{1:T} \mid s, a_0) \right).$$

According to the Jensen's Inequality (convexity of $-\log$) and $Q(s, a) \geq 0$, the second KL divergence term

$$
\begin{aligned}
& Q(s, a_0) \, \mathrm{D}_{KL} \left( q (a_{1:T} \mid s, a_0) \, \| \, \pi_\theta (a_{1:T} \mid s, a_0) \right) \\
&= Q(s, a_0) \mathbb{E}_{a_{1:T} \sim q(a_{1:T}|s, a_0)} \left[ \log \frac{q (a_{1:T} \mid s, a_0)}{\pi_\theta (a_{1:T} \mid s, a_0)} \right] \\
&= Q(s, a_0) \mathbb{E}_{a_{1:T} \sim q(a_{1:T}|s, a_0)} \left[ -\log \frac{\pi_\theta (a_{1:T} \mid s, a_0)}{q (a_{1:T} \mid s, a_0)} \right] \\
&\geq -Q(s, a_0) \log \left( \mathbb{E}_{a_{1:T} \sim q(a_{1:T}|s, a_0)} \left[ \frac{\pi_\theta (a_{1:T} \mid s, a_0)}{q (a_{1:T} \mid s, a_0)} \right] \right) \\
&= -Q(s, a_0) \log(1) \\
&= 0.
\end{aligned}
$$

Obviously, the equality holds when $\pi_\theta (a_{1:T} \mid s, a_0) = q (a_{1:T} \mid s, a_0)$. Finally, the Q-weighted variational bound objective

$$\mathbb{E}_{s, a_0 \sim p(s'|s, a_0), \pi_k(a|s)} \left[ Q(s, a_0) \mathbb{E}_{a_{1:T} \sim q(a_{1:T}|s, a_0)} \left[ \log \frac{\pi_\theta (a_{0:T} \mid s)}{q (a_{1:T} \mid s, a_0)} \right] \right]$$

is the tight lower bound of the RL policy objective

$$\mathbb{E}_{s, a \sim p(s'|s,a), \pi_k(a|s)} \left[ Q(s, a) \log(\pi_\theta(a|s)) \right]$$

$\square$

## A.2 Deriviation of Q-weighted VLO Loss

For each state $s$, we have

$$\mathbb{E}_{a_{1:T}\sim q(a_{1:T}|s,a_0)}\left[\log\frac{q\left(a_{1:T}\mid a_0\right)}{\pi_\theta\left(a_{0:T}\right)}\right]$$

$$= \mathbb{E}_{a_{1:T}\sim q(a_{1:T}|s,a_0)}\left[\log\frac{\prod_{t=1}^{T}q\left(a_t\mid a_{t-1}\right)}{\pi_\theta\left(a_T\right)\prod_{t=1}^{T}\pi_\theta\left(a_{t-1}\mid a_t\right)}\right]$$

$$= \mathbb{E}_{a_{1:T}\sim q(a_{1:T}|s,a_0)}\left[-\log\pi_\theta\left(a_T\right)+\log\frac{\prod_{t=1}^{T}q\left(a_t\mid a_{t-1}\right)}{\prod_{t=1}^{T}\pi_\theta\left(a_{t-1}\mid a_t\right)}\right]$$

$$= \mathbb{E}_{a_{1:T}\sim q(a_{1:T}|s,a_0)}\left[-\log\pi_\theta\left(a_T\right)+\sum_{t=1}^{T}\log\frac{q\left(a_t\mid a_{t-1}\right)}{\pi_\theta\left(a_{t-1}\mid a_t\right)}\right]$$

$$= \mathbb{E}_{a_{1:T}\sim q(a_{1:T}|s,a_0)}\left[-\log\pi_\theta\left(a_T\right)+\sum_{t=2}^{T}\log\frac{q\left(a_t\mid a_{t-1}\right)}{\pi_\theta\left(a_{t-1}\mid a_t\right)}+\log\frac{q\left(a_1\mid a_0\right)}{\pi_\theta\left(a_0\mid a_1\right)}\right]$$

$$= \mathbb{E}_{a_{1:T}\sim q(a_{1:T}|s,a_0)}\left[-\log\pi_\theta\left(a_T\right)+\sum_{t=2}^{T}\log\frac{q\left(a_{t-1}\mid a_t,a_0\right)q\left(a_t\mid a_0\right)}{\pi_\theta\left(a_{t-1}\mid a_0\right)q\left(a_{t-1}\mid a_0\right)}+\log\frac{q\left(a_1\mid a_0\right)}{\pi_\theta\left(a_0\mid a_1\right)}\right]$$

$$= \mathbb{E}_{a_{1:T}\sim q(a_{1:T}|s,a_0)}\left[-\log\pi_\theta\left(a_T\right)+\sum_{t=2}^{T}\log\left(\frac{q\left(a_{t-1}\mid a_t,a_0\right)}{\pi_\theta\left(a_{t-1}\mid a_t\right)}\cdot\frac{q\left(a_t\mid a_0\right)}{q\left(a_{t-1}\mid a_0\right)}\right)+\log\frac{q\left(a_1\mid a_0\right)}{\pi_\theta\left(a_0\mid a_1\right)}\right]$$

$$= \mathbb{E}_{a_{1:T}\sim q(a_{1:T}|s,a_0)}\left[-\log\pi_\theta\left(a_T\right)+\sum_{t=2}^{T}\left(\log\frac{q\left(a_{t-1}\mid a_t,a_0\right)}{\pi_\theta\left(a_{t-1}\mid a_t\right)}+\log\frac{q\left(a_t\mid a_0\right)}{q\left(a_{t-1}\mid a_0\right)}\right)+\log\frac{q\left(a_1\mid a_0\right)}{\pi_\theta\left(a_0\mid a_1\right)}\right]$$

$$= \mathbb{E}_{a_{1:T}\sim q(a_{1:T}|s,a_0)}\left[-\log\pi_\theta\left(a_T\right)+\sum_{t=2}^{T}\log\frac{q\left(a_{t-1}\mid a_t,a_0\right)}{\pi_\theta\left(a_{t-1}\mid a_t\right)}+\log\frac{q\left(a_T\mid a_0\right)}{q\left(a_1\mid a_0\right)}+\log\frac{q\left(a_1\mid a_0\right)}{\pi_\theta\left(a_0\mid a_1\right)}\right]$$

$$= \mathbb{E}_{a_{1:T}\sim q(a_{1:T}|s,a_0)}\left[-\log\pi_\theta\left(a_T\right)+\sum_{t=2}^{T}\log\frac{q\left(a_{t-1}\mid a_t,a_0\right)}{\pi_\theta\left(a_{t-1}\mid a_t\right)}+\log q\left(a_T\mid a_0\right)\right.$$

$$\left.-\log q\left(a_1\mid a_0\right)+\log q\left(a_1\mid a_0\right)-\log\pi_\theta\left(a_0\mid a_1\right)\right]$$

$$= \mathbb{E}_{a_{1:T}\sim q(a_{1:T}|s,a_0)}\left[\log\frac{q\left(a_T\mid a_0\right)}{\pi_\theta\left(a_T\right)}+\sum_{t=2}^{T}\log\frac{q\left(a_{t-1}\mid a_t,a_0\right)}{\pi_\theta\left(a_{t-1}\mid a_t\right)}-\log\pi_\theta\left(a_0\mid a_1\right)\right]$$

$$= \mathbb{E}_{a_{1:T}\sim q(a_{1:T}|s,a_0)}\left[D_{KL}\left(q\left(a_T\mid a_0\right)\|\pi_\theta\left(a_T\right)\right)+\sum_{t=2}^{T}D_{KL}\left(q\left(a_{t-1}\mid a_t,a_0\right)\|\pi_\theta\left(a_{t-1}\mid a_t\right)\right)-\log\pi_\theta\left(a_0\mid a_1\right)\right]$$

$$= \mathbb{E}_{a_0,\epsilon}\left[\frac{\beta_t^2}{2\sigma_t^2\alpha_t\left(1-\bar\alpha_t\right)}\left\|\epsilon-\epsilon_\theta\left(\sqrt{\bar\alpha_t}a_0+\sqrt{1-\bar\alpha_t}\epsilon,s,t\right)\right\|^2\right].$$

Thus, the Q-weighted VLO loss can be derived as

$$\max_\theta\mathbb{E}_{s,a_0\sim\pi_k(a|s)}\left[Q(s,a_0)\mathbb{E}_{a_{1:T}\sim q(a_{1:T}|s,a_0)}\left[\log\frac{\pi_\theta\left(a_{0:T}\mid s\right)}{q\left(a_{1:T}\mid s,a_0\right)}\right]\right]$$

$$\triangleq\min_\theta\mathbb{E}_{s,a_0\sim\pi_k(a|s)}\left[Q(s,a_0)\mathbb{E}_{a_{1:T}\sim q(a_{1:T}|s,a_0)}\left[\log\frac{q\left(a_{1:T}\mid s,a_0\right)}{\pi_\theta\left(a_{0:T}\mid s\right)}\right]\right]$$

$$\triangleq\min_\theta\mathbb{E}_{s,a\sim\pi_k(a|s),\epsilon,t}\left[\frac{\beta_t^2}{2\sigma_t^2\alpha_t\left(1-\bar\alpha_t\right)}Q(s,a)\cdot\left\|\epsilon-\epsilon_\theta\left(\sqrt{\bar\alpha_t}a+\sqrt{1-\bar\alpha_t}\epsilon,s,t\right)\right\|^2\right].$$

## A.3 Proof of Theorem 2

**Theorem 2.** *(Optimal Solution in One Policy Improvement Step)* *Considering the optimization problem in policy improvement step* $\pi_{k+1}=\arg\max_\pi\mathbb{E}_{\tau\sim\pi_k}[Q_{\pi_k}(s,a)\log(\pi)]$, *the optimal policy*

*in a given state $s$ is*

$$p(a \mid s) = \begin{cases} \frac{\pi(a|s)Q(s,a)}{\int \mathbb{I}_{Q(s,a)>0}(a)\pi(a|s)Q(s,a)da}, & Q(s,a) > 0 \\ 0, & otherwise \end{cases} \tag{11}$$

*when there exists $a$ such that $Q(s,a) > 0$, and*

$$p(a \mid s) = \frac{1}{N}\sum_{i=1}^{N}\delta(x - a_i), a_i \in \left\{a \mid Q(s,a) = \max_a Q(s,a)\right\}, \tag{12}$$

*when $Q(s,a) \leq 0$ for any $a \in \mathcal{A}$, $\delta(x - a)$ is the Dirac function, $N$ is the cardinality of set $\{a \mid Q(s,a) = \max_a Q(s,a)\}$, $\pi(a \mid s)$ is the behavior policy, $p(a \mid s)$ is the policy to be optimized and $\mathbb{I}_{Q(s,a)>0}$ is the indicator function that judges whether $Q(s,a) > 0$.*

*Proof.* Consider the optimization problem of the policy improvement step in a certain state $s$:

$$\min_{p(a|s)} -\int_{\mathcal{A}} \pi(a \mid s)Q(s,a)\log p(a \mid s)da$$

$$\text{subject to } \int p(a \mid s)da = 1$$

$$p(a \mid s) \geq 0, a \in \mathcal{A}$$

where $p(a \mid s)$ is a probability density function defined on $\mathcal{A}$ that represents the policy to be optimized in the state $s$.

Here, we can derive the solution to this optimization problem according to KKT conditions by classifying it into three cases.

- $Q(s,a) > 0$ for any action $a \in \mathcal{A}$. In this case, the optimization problem is convex, and we can use KKT conditions directly to obtain its optimal solution as below:

$$\frac{\pi(a \mid s)Q(s,a)}{p(a \mid s)} - \lambda + \nu(a \mid s) = 0$$

$$\int p(a \mid s)da = 1 \tag{13}$$

$$\nu(a \mid s) \geq 0, a \in \mathcal{A}$$

$$\nu(a \mid s)p(a \mid s) = 0, a \in \mathcal{A}$$

  Then, consider

$$p(a|s) = \frac{\pi(a \mid s)Q(s,a)}{\lambda - \nu(a|s)}.$$

  Since $Q(s,a) > 0$ and $p(a|s) \geq 0$, the denominator

$$\lambda - \nu(a \mid s) \geq 0.$$

  Hence,

$$p(a \mid s) > 0, \quad \nu(a \mid s) = 0,$$

  which means

$$p(a \mid s) = \frac{\pi(a \mid s)Q(s,a)}{\int \pi(a \mid s)Q(s,a)da}.$$

- $Q(s,a) \leq 0$ for any action $a \in \mathcal{A}$. When $Q(s,a) \leq 0$ for any action $a \in \mathcal{A}$, it can be found that the objective will be $-\infty$ if there exists an action $a$ such that $p(a|s) = 0$ and $\pi(a|s)Q(a \mid s) < 0$. In that case, only considering the original optimization problem is not meaningful. Hence, we can further consider the optimality of $\int Q(s,a)p(a|s)da$. Finally, we can drive the optimal policy pdf is the summation of the Dirac function

$$p(a \mid s) = \frac{1}{N}\sum_{i=1}^{N}\delta(x - a_i), a_i \in \left\{a \mid Q(s,a) = \max_a Q(s,a), \pi(a \mid s) > 0\right\}$$

  where $N$ is the cardinality of set $\{a \mid Q(s,a) = \max_a Q(s,a), \pi(a \mid s)\}$.

- $Q(s, a) > 0$ for some actions $a$ and $Q(s, a) \leq 0$ for the left actions $a$. Combining the above two situations, the optimal policy pdf is

$$p(a \mid s) = \begin{cases} \frac{\pi(a|s)Q(s,a)}{\int \mathbb{I}_{Q(s,a)>0}\pi(a|s)Q(s,a)da} & Q(s,a) > 0 \\ 0 & otherwise \end{cases}.$$

Finally, we can summarize the optimal policy in a certain state $s$ as

$$p(a \mid s) = \begin{cases} \frac{\pi(a|s)Q(s,a)}{\int \mathbb{I}_{Q(s,a)>0}(a)\pi(a|s)Q(s,a)da}, & Q(s,a) > 0 \\ 0, & otherwise \end{cases} \tag{14}$$

when there exists $a$ such that $Q(s, a) > 0$, and

$$p(a \mid s) = \frac{1}{N}\sum_{i=1}^{N}\delta(x - a_i), a_i \in \left\{ a \mid Q(s,a) = \max_a Q(s,a), \pi(a \mid s) > 0 \right\}, \tag{15}$$

when $Q(s, a) \leq 0$ for any $a \in \mathcal{A}$. $\qquad\square$

### A.4 Proof for Convergence of QVPO

Assume the new diffusion policy after one QVPO iteration can be approximately expressed as

$$\pi_{new}(a \mid s) \approx (1 - p_{data}(s)A_{\pi_{old}}(s, a^\star)\eta)\pi_{old}(a \mid s) + p_{data}(s)A_{\pi_{old}}(s, a^\star)\eta\frac{\mathbb{I}_{a\in\mathcal{N}(a^\star|s,\epsilon)}(a)}{S_{\mathcal{N}(a^\star|s,\epsilon)}}$$

where $a^\star \mid s$ is the action that can maximize $Q_{\pi_{old}}(s, a)$ in the state $s$, $\mathcal{N}(a^\star \mid s, \epsilon)$ is the neighborhood of $a^\star \mid s$ with a small radius $\epsilon$, $S_{\mathcal{N}(a^\star|s,\epsilon)}$ is the area of this neighborhood, and $p_{data}(s)$ is the sampling distribution of the state. This assumption is straightforward: the training sample's generation probability in the diffusion model will be improved and the improved probability is proportional to its weight.

Now consider the improvement of the RL objective:

$$\begin{aligned}
\mathcal{J}(\pi_{new}) - \mathcal{J}(\pi_{old}) &= \mathbb{E}_{s\sim\rho_0}\left[V_{\pi_{new}}(s) - V_{\pi_{old}}(s)\right] \\
&= \mathbb{E}_{s\sim\rho_0}\left[\mathbb{E}_{a\sim\pi_{new}(a|s)}\left[Q_{\pi_{new}}(s, a)\right] - V_{\pi_{old}}(s)\right] \\
&= \mathbb{E}_{s\sim\rho_0}\left[\mathbb{E}_{a\sim\pi_{new}(a|s)}\left[Q_{\pi_{new}}(s, a) - Q_{\pi_{old}}(s, a)\right]\right] \\
&\quad + \mathbb{E}_{s\sim\rho_0}\left[\mathbb{E}_{a\sim\pi_{new}(a|s)}\left[Q_{\pi_{old}}(s, a)\right] - V_{\pi_{old}}(s)\right] \\
&= \mathbb{E}_{s\sim\rho_0}\left[\mathbb{E}_{a\sim\pi_{new}(a|s)}\left[Q_{\pi_{new}}(s, a) - Q_{\pi_{old}}(s, a)\right]\right] + \mathbb{E}_{s,a\sim\rho_0,\pi_{new}(a|s)}\left[A_{\pi_{old}}(s, a)\right]
\end{aligned}$$

The first term here can be further expanded according to the Bellman equation:

$$\mathbb{E}_{s\sim\rho_0}\left[\mathbb{E}_{a\sim\pi_{new}(a|s)}\left[Q_{\pi_{new}}(s, a) - Q_{\pi_{old}}(s, a)\right]\right] = \gamma\mathbb{E}_{s\sim d^1_{\pi_{new}}}\left[V_{\pi_{new}}(s) - V_{\pi_{old}}(s)\right]$$

where $d^1_{\pi_{new}}$ denotes the probability distribution of state in time step $t = 1$ with policy $\pi_{new}$. Repeating the above operation, we will obtain:

$$\begin{aligned}
\mathcal{J}(\pi_{new}) - \mathcal{J}(\pi_{old}) &= \sum_{t=0}^{\infty}\gamma^t\mathbb{E}_{s,a\sim d^t_{\pi_{new}},\pi_{new}}\left[A_{\pi_{old}}(s, a)\right] \\
&= \frac{1}{1-\gamma}\mathbb{E}_{s\sim d_{\pi_{new}}}\left[\mathbb{E}_{a\sim\pi_{new}(\cdot|s)}\left[A_{\pi_{old}}(s, a)\right]\right] \\
&\approx \frac{1}{1-\gamma}\mathbb{E}_{s\sim d_{\pi_{new}}}\left[(1 - p_{data}(s)A_{\pi_{old}}(s, a^\star)\eta)\mathbb{E}_{a\sim\pi_{old}(\cdot|s)}\left[A_{\pi_{old}}(s, a)\right]\right. \\
&\quad \left. + p_{data}(s)A^2_{\pi_{old}}(s, a^\star)\eta\frac{\mathbb{I}_{a\in\mathcal{N}(a^\star|s,\epsilon)}(a)}{S_{\mathcal{N}(a^\star|s,\epsilon)}}\right] \\
&= \frac{1}{1-\gamma}\mathbb{E}_{s\sim d_{\pi_{new}}}\left[p_{data}(s)A^2_{\pi_{old}}(s, a^\star)\eta\frac{\mathbb{I}_{a\in\mathcal{N}(a^\star|s,\epsilon)}(a)}{S_{\mathcal{N}(a^\star|s,\epsilon)}}\right] \geq 0
\end{aligned}$$

# B  Hyperparameters

All of our experiments are implemented on a GPU of NVIDIA GeForce RTX 4090 with 24GB and a CPU of Intel Xeon w5-3435X. The implementation of SAC, TD3, PPO, SPO, and DIPO is based on `https://github.com/toshikwa/soft-actor-critic.pytorch`, `https://github.com/sfujim/TD3`, `https://github.com/ikostrikov/pytorch-a2c-ppo-acktr-gail`, `https://github.com/MyRepositories-hub/Simple-Policy-Optimization`, and `https://github.com/BellmanTimeHut/DIPO` respectively, which are official code library. Table 2 and 3 present the hyper-parameters used in our experiments. Notably, we find that the performance of DIPO with 20 diffusion steps is worse. Thus, different from 20 diffusion steps in QVPO, 100 diffusion steps are set for DIPO, which are also recommended in the original paper [42].

Table 2: Hyper-parameters used in the experiments.

| Hyperparameters | QVPO | DIPO | SAC | TD3 | SPO | PPO |
|---|---|---|---|---|---|---|
| No. of hidden layers | 2 | 2 | 2 | 2 | 2 | 2 |
| No. of hidden nodes | 256 | 256 | 256 | 256 | 64 | 256 |
| Activation | mish | mish | relu | relu | tanh | tanh |
| Batch size | 256 | 256 | 256 | 256 | 256 | 256 |
| Discount for reward $\gamma$ | 0.99 | 0.99 | 0.99 | 0.99 | 0.99 | 0.99 |
| Target smoothing coefficient $\tau$ | 0.005 | 0.005 | 0.005 | 0.005 | 0.0 | 0.005 |
| Learning rate for actor | $3 \times 10^{-4}$ | $3 \times 10^{-4}$ | $3 \times 10^{-4}$ | $3 \times 10^{-4}$ | $3 \times 10^{-4}$ | $7 \times 10^{-4}$ |
| Learning rate for critic | $3 \times 10^{-4}$ | $3 \times 10^{-4}$ | $3 \times 10^{-4}$ | $3 \times 10^{-4}$ | $3 \times 10^{-4}$ | $7 \times 10^{-4}$ |
| Actor Critic grad norm | N/A | 2 | N/A | N/A | 0.5 | 0.5 |
| Memory size | $1 \times 10^{6}$ | $1 \times 10^{6}$ | $1 \times 10^{6}$ | $1 \times 10^{6}$ | $1 \times 10^{6}$ | $1 \times 10^{6}$ |
| Entropy coefficient | N/A | N/A | 0.2 | N/A | N/A | 0.01 |
| Value loss coefficient | N/A | N/A | N/A | N/A | N/A | 0.5 |
| Exploration noise | N/A | N/A | N/A | $\mathcal{N}(0, 0.1)$ | N/A | N/A |
| Policy noise | N/A | N/A | N/A | $\mathcal{N}(0, 0.2)$ | N/A | N/A |
| Noise clip | N/A | N/A | N/A | 0.5 | N/A | N/A |
| Use gae | N/A | N/A | N/A | N/A | True | True |
| Diffusion steps | 20 | 100 | N/A | N/A | N/A | N/A |

Table 3: Hyper-parameters used in QVPO.

| Hyperparameters | Ant-v3 | HalfCheetah-v3 | Hopper-v3 | Humanoid-v3 | Waller2d-v3 |
|---|---|---|---|---|---|
| Number of samples from diffusion policy $N_d$ 64 | 64 | 64 | 64 | 64 | 64 |
| Number of samples from $\mathcal{U}(\underline{a}, \overline{a})$ $N_e$ | 10 | 10 | 10 | 10 | 10 |
| Action selection number for behavior policy $K_b$ | 4 | 4 | 4 | 4 | 4 |
| Action selection number for target policy $K_t$ | 2 | 4 | 1 | 2 | 2 |
| Entropy weight $\omega_{ent}$ | 0.01 | 0.01 | 0.01 | 0.01 | 0.01 |
| Q-weight transformation function | $qadv$ | $qadv$ | $qadv$ | $qadv$ | $qadv$ |

# C  Training and Inference Time

The comparison of training and inference time is shown in the following tables. Notably, since the official implementation of QSM is based on the jax and other algorithms are based on pytorch, it is not a fair comparison. In practice, the algorithm realized with jax is 6-10 times faster than that realized with pytorch. Besides, to fairly compare the diffusion-based RL methods in training and inference time, we set the same number of diffusion steps (T=20) for all of them (i.e., QVPO, QSM, and DIPO). The results in Table 4 imply that QVPO can sufficiently use the parallel computing ability of GPU and the multiple-sampling and selecting procedure does not take much time compared with gradient-based optimization like DIPO.

Table 4: The training and inference time comparison on Ant-v3 Benchmarks.

| Method | QVPO | DIPO | TD3 | SAC | PPO | SPO | QSM (jax) |
|---|---|---|---|---|---|---|---|
| Training Time (h) | 6.8 | 10.5 | 0.5 | 2.5 | 0.3 | 0.3 | 1.0 |
| Inference Time (ms) | 6.2 | 5.7 | 0.2 | 0.3 | 0.2 | 0.3 | 0.9 |

Moreover, although QVPO is much slower than classical RL methods like SAC in inference, the inference time (6ms) still is acceptable. To our knowledge, most existing real robots only require 50-Hz control policy (i.e. output action per 20 ms). Besides, just like QSM, the inference time of QVPO can be further improved with jax framework if it is necessary. Hence, the inference time is not a bottleneck to applying QVPO to real applications.

