# OpenReview forum: "Diffusion-based Reinforcement Learning via Q-weighted Variational Policy Optimization"
_NeurIPS.cc/2024/Conference — NeurIPS 2024 poster_

### Official Review · Reviewer_uSjP · 2024-07-10

**Soundness:** 3
**Presentation:** 4
**Contribution:** 2
**Rating:** 5
**Confidence:** 4

**Summary:**

This paper introduces QVPO, a new model-free algorithm that trains a diffusion policy online. It proposes a Q-weighted VLO loss by weighting the original diffusion model objective with Q-values. To address the issue of negative Q-values, the algorithm uses advantage instead. QVPO also encourages exploration by mimicking the uniform distribution and reduces inference variance by executing actions with higher Q-values.

**Strengths:**

- The paper addresses a relevant problem, which is to learn diffusion policy online.

- The paper is generally well-written and easy to read.

- The proposed method demonstrates impressive results on MuJoCo benchmarks compared to SOTA baselines.

**Weaknesses:**

- Several claims require additional support. For instance, while Figure 2 is informative in demonstrating the ideal effect of the entropy term, experimental results on toy examples would provide stronger evidence than an illustration.

- Since QVPO relies on action samples for both optimization and action selection, this will increase computational demands.

- The tasks are limited and relatively trivial, making it difficult to tell performance in more complex tasks, such as robot manipulation. In particular, when the action space is larger, it is unclear how the entropy term will scale given its reliance on uniform action distributions.

**Questions:**

- Can the author provide an explanation for why the exploration ability of diffusion policies declines with a limited number of diffusion steps? Additionally, what happens when the diffusion steps are even fewer, say 5, a common value in other diffusion policy literature? I suppose the entropy term should perform better in such cases.

- If the maximum entropy term is used throughout the training, the policy may continue to follow the uniform distribution in the later stages, potentially hurting performance and convergence. If this is the case, would a scheduling mechanism improve the final performance?

- Can the author explain why the Q-weighted VLO loss is superior to QSM regarding the additional errors in policy optimization?

- In line 300, it is mentioned that action selection is used during inference, which is a standard technique in offline RL. However, is the same technique applied to the baselines? If not, this would be unfair.

**Limitations:**

--

---

> ### Author Rebuttal · Authors · 2024-08-07
>
> We thank the reviewer for your valuable feedback and comments! We itemize the weaknesses you mentioned and answer them.
>
> > **Q1**: While Figure 2 is informative in demonstrating the ideal effect of the entropy term, experimental results on toy examples would provide stronger evidence than an illustration.
>
> **A1**: We perform a toy experiment on a continuous bandit problem to further present the effect of the entropy term. The results are shown in Figure 1 of the attached pdf file, which is similar to the illustration. We will add it in the final version.
>
> > **Q2**: Since QVPO relies on action samples for both optimization and action selection, this will increase computational demands.
>
> **A2**: Actually, that is not the weakness but the advantage of QVPO. QVPO can sufficiently utilize the parallel computing ability of GPU via the multiple sampling and selection procedure. We observed that the previous diffusion-based RL methods generally have a low gpu utilization rate since the computational bottleneck of the forward process in the diffusion. In that case, parallelly sampling multiple samples does not add much extra time cost.
>
> Besides, QVPO does not need to use the gradient from Q network to optimize the diffusion policy. This also reduces the time cost to a certain extent compared with other diffusion RL methods like DIPO. Finally, the comprehensive training time and inference time comparison can be referred to in **Global Rebuttal**.
>
> > **Q3**: The tasks are limited and relatively trivial, making it difficult to tell performance in more complex tasks, such as robot manipulation. In particular, when the action space is larger, it is unclear how the entropy term will scale given its reliance on uniform action distributions.
>
> **A3**: We need to clarify that these 5 mujoco environments are the standard online RL benchmarks and most of the previous online RL methods (e.g., SAC, DIPO, etc.) only choose them to do the experiments. That is why we verify our algorithm on them.
>
> Besides, to further present the efficiency of QVPO, we also apply it to the recently released HumanoidBench environment, which is a complex continuous control benchmark based on Unitree H1 humanoid with **151 observation dimension** and **61 action dimension**. The results are shown in Figure 3 of the attached pdf file. It can be observed that model-free RL methods (SAC, PPO) do not work totally, while QVPO (as a model-free method) achieves superior performance compared with advanced model-based methods (Dreamer-v3, TD-MPC2).
>
> The entropy coefficient is set 0.01 here as well. In fact, we found that our algorithm is robust enough with the scaling of $w_{ent}$ in 1e-2 to 1e-3.
>
> > **Q4(1)**: Can the author provide an explanation for why the exploration ability of diffusion policies declines with a limited number of diffusion steps?
>
> **A4(1)**: Note that the Gaussian noise is added at each diffusion denoising step. In other words, more diffusion steps lead to more possibilities. Thus, the exploration ability of diffusion policies declines with a limited number of diffusion steps.
>
> > **Q4(2)**: Additionally, what happens when the diffusion steps are even fewer, say 5, a common value in other diffusion policy literature? I suppose the entropy term should perform better in such cases.
>
> **A4(2)**: We add the result of the ablation study with 5 diffusion steps in the attached pdf file. It can be found that the performance is poor. Hence, different from offline RL methods, a certain number of diffusion steps is required in online RL setting. This is also verified in previous works like DIPO [R7].
>
> > **Q5**: If the maximum entropy term is used throughout the training, the policy may continue ... potentially hurting performance and convergence. Would a scheduling mechanism improve the final performance?
>
> **A5**: The entropy term is not constant as you thought. As mentioned in Algorithm 1, line 8. You can find that the actual entropy coefficient $w_{ent}(s)=w_{ent}w_{eq}(s,a_{max})$ is related to $w_{eq}(s,a_{max})=A(s,a_{max}), A(s,a_{max})\ge 0$. When QVPO converges, $A(s,a_{max})\rightarrow 0$ and $w_{ent}(s)\rightarrow 0$ .
>
> > **Q6**: Can the author explain why the Q-weighted VLO loss is superior to QSM regarding the additional errors in policy optimization?
>
> **A6**: As mentioned in lines 43-46, QSM has a doubled approximation error from the alignment process. More concretely, the trained Q value function has an approximation error. QSM utilizes the gradient of this approximated Q function to train the score model. This leads to the doubled approximation error in the score model. That is why the performance of QSM is worse than QVPO as shown in the attached pdf. Besides, the policy improvement stage of DIPO is sampling-based, which implies QVPO is more likely to cross the local optimum.
>
> > **Q7**: In line 300, it is mentioned that action selection is used during inference, which is a standard technique in offline RL. However, is the same technique applied to the baselines? If not, this would be unfair.
>
> **A7**: We need to clarify that there is a contradiction if we apply this technique to other methods.
>
> As mentioned in offline RL works like IDQL [R6], applying action selection to diffusion policy is to yield a deterministic policy during inference. For MLP or Gaussian policy, we can directly output the deterministic action or the mean of Gaussian policy to achieve this goal. Hence, this trick is not appropriate for classical online RL methods (e.g., SAC, TD3).
>
> For diffusion-based RL, like DIPO, it yields the deterministic policy via fixing the denoising process (i.e., updating with the mean value in each diffusion step). In that case, there exists a contradiction if action selection is also applied during inference.
>
> [R6] Hansen P, et al. Idql: Implicit q-learning as an actor-critic method with diffusion policies.
>
> [R7] Yang L, et al. Policy representation via diffusion probability model for reinforcement learning.

---

> > ### Comment · Reviewer_uSjP · 2024-08-10
> >
> > I appreciate the response and the effort made by the authors during the rebuttal. However, I still have a couple of questions. First,  the result of continuous bandit problem looks perfect. Can you elaborate more on how the experiment is actually being done, e.g., how do you sample the initial points? Second, the authors claim that QVPO has a high GPU utilization compared to baselines. However, do you use any parallel environment during training? I believe that parallel environments, either on CPU or GPU, have been a common technique in RL training. If you use hundreds of parallel environments, say 256 (a common number that most devices can support), the GPU utilization will be high for baselines too, and therefore the multiple sampling and selection procedure of QVPO will become a bottleneck, let alone GPU simulators with thousands of parallel environments. Third, the authors claim that QVPO can leverage the multimodality of diffusion policies (line 60), however, the authors didn't prove it or even mention it in the rest of the paper. I feel like this claim is very strong and you should verify it in experiments.

---

> > > ### Author Response · Authors · 2024-08-12
> > >
> > > Thanks for your comments and questions as well.
> > >
> > > **Q8**: Can the authors elaborate more on how the continuous bandit experiment is actually being done, e.g., how do you sample the initial points?
> > >
> > > **A8**: The continuous bandit experiment is done similarly to Algorithm 1. Concretely, we sample 64 actions from the current diffusion policy and select the best according to the reward function $Q(x)=\sum_{i=1}^3w_i\frac1{2\pi\sigma_i^2}\exp\left(-\frac1{2\sigma_i}(x-\mu_i)^T(x-\mu_i)\right)$, where $w_i=1.5$, $\sigma_i=0.1$, and $\mu_i=[-1.35,0.65]^T$, $[-0.65,1.35]^T$ and $[-1.61,1.61]^T$ respectively. Then, we train the diffusion policy with the best action sample (with & without entropy term) in each training epoch. We plot the evaluation results per 10 training epochs with enough samples from the diffusion policy.
> > >
> > > The initial points (we guess you mean the first column of Figure 1 in the attached pdf file) are sampled from an initial diffusion policy pre-trained from a Gaussian distribution $\mathcal{N}(0, 0.25)$. Notably, **this pre-training procedure is not required in practical implementation**. The pre-training procedure from a given Gaussian distribution (not good) actually increases the convergence difficulty for QVPO. We use this pre-training procedure just for better visualization; otherwise, QVPO will converge very fast for such a toy example and thus cannot show a good visualization.
> > >
> > > **Q9**: Do you use any parallel environment during training? The multiple sampling and selection procedure of QVPO may become a bottleneck, let alone GPU simulators with thousands of parallel environments.
> > >
> > > **A9**: No, we do not use any parallel environment during training for fair comparison. Firstly, we need to clarify that the parallelism of QVPO is not in **the online environment interaction stage** but **in the policy optimization stage**. As mentioned in lines 178-179, the difficulty of diffusion policy optimization is how to obtain the optimal action samples. Previous diffusion-based RL method like DIPO finds the optimal action sample by performing multiple gradient updates on the actions in the replay buffer. However, the multiple gradient updates in DIPO cannot be implemented in parallel. In contrast, QVPO finds the optimal action samples via multiple sampling and selection, which can be parallelly executed. In that case, QVPO can speed up policy optimization compared to DIPO.
> > >
> > > Besides, although the idea of action selection is also applied to online environment interaction (i.e., "efficient diffusion policy" in Section 4.4), we believe this trick of QVPO is not a bottleneck with parallel environment interaction. This is because the action selection number in "efficient diffusion policy" is commonly chosen to be very small ($K_b=4$ is enough generally). In that case, it will not affect the training time too much with parallel environments. In fact, the GPU utilization and memory consumption of QVPO is limited (i.e., 55% & 598MB) even in the policy optimization stage, where 256 states are given parallelly and we need to sample 64 actions for each of them.
> > >
> > > **Q10**: The authors claim that QVPO can **leverage** the multimodality of diffusion policies (line 60), however, the authors didn't prove it or even mention it in the rest of the paper. I feel like this claim is very strong and you should verify it in experiments.
> > >
> > > **A10**: We have no intention to show the multimodality of diffusion policy (which has been proven in DIPO, QSM). Instead, we want to efficiently **leverage** the multimodality of diffusion policy to achieve better performance in continuous control tasks. The multimodality is generally hard to visualize, especially for complex real tasks. Therefore, we develop a toy example to show this property. As shown in Figure 1 of the attached file, the proposed diffusion entropy term and the sampling-based policy optimization make it easy for QVPO to explore broader action space (Figure 1) and thus can verify that QVPO can better leverage multimodality.

---

> > > > ### Comment · Reviewer_uSjP · 2024-08-12
> > > >
> > > > Thanks for the response. Regarding A8, I am curious what if the rewards for each center are unbalanced and have different weights? Overall, I think the rebuttal alleviates my concerns and I will raise my score to 5 (slightly beyond the accept line).

---

> > > > > ### Author Response · Authors · 2024-08-13
> > > > >
> > > > > **Q11**: What if the rewards for each center are unbalanced and have different weights?
> > > > >
> > > > > **A11**: Thanks for raising the score. That is a good question. Actually, We conducted the experiments under this setting before. We find that QVPO with the entropy term finally converges to the optimal center under an unbalanced center setting, while QVPO without the entropy term has a certain probability of falling into the sub-optimal center, especially in the cases where the imbalance between different modes is not obvious. Concretely, QVPO with entropy term usually first explores these three centers (show its multimodality) and then converges to the optimal one. The convergence rate depends on the imbalance degree (i.e., the greater the relative weight difference among the three modes, the faster the convergence).

---

### Official Review · Reviewer_Qp9e · 2024-07-12

**Soundness:** 3
**Presentation:** 2
**Contribution:** 3
**Rating:** 5
**Confidence:** 4

**Summary:**

The paper proposes a novel model-free online reinforcement learning (RL) algorithm called Q-weighted Variational Policy Optimization (QVPO), which leverages the expressiveness and multimodality of diffusion models. By introducing Q-weighted variational loss and entropy regularization, the authors aim to overcome the limitations of unimodal policies and enhance exploration capabilities. The algorithm is validated through experiments on MuJoCo continuous control benchmarks, demonstrating state-of-the-art performance in terms of both cumulative reward and sample efficiency.

**Strengths:**

The paper presents a significant advancement in integrating diffusion models into online RL, a relatively underexplored area. The proposed QVPO algorithm is theoretically sound, with the Q-weighted variational loss being proven as a tight lower bound of the policy objective in online RL. The incorporation of an entropy regularization term and an efficient behavior policy are innovative approaches to enhance exploration and sample efficiency.

**Weaknesses:**

1. Experiment Scope: The experiments are limited to D4RL datasets, which do not cover the full spectrum of possible RL environments. The generalization of the proposed method to other types of tasks, such as discrete action spaces or tasks with sparse rewards, is not demonstrated, eg. Adroit, Humanoid, MetaWorld, Maniskill, KUKA Pick-and-Place.
2. Ablation Studies: While the paper includes ablation studies, they are not exhaustive. Additional ablation studies are needed to isolate the contributions of individual components, such as the Q-weighted variational loss and the entropy regularization term. It is unclear how much each component independently contributes to the overall performance improvements.
3. Diffusion Policy Variance Reduction: The proposed method for reducing diffusion policy variance via action selection aims to improve sample efficiency. However, this approach may sacrifice the exploration capabilities of the diffusion policy, thus limiting its multimodal potential. This trade-off is not discussed in the paper, and no ablation study is provided to evaluate the impact.

**Questions:**

1. Can the proposed algorithm be adapted for tasks with discrete action spaces or sparse rewards, and if so, how?
2. How is the diffusion policy initialized, and what impact does this initialization have on the performance of the algorithm? If the policy is tranined from scratch, how is the Q-function trained at the beginning stage?
3. The efficient diffusion policy method reduces policy variance but may limit exploration. How do you balance exploration and exploitation, and can you provide an ablation study to verify the effects of this trade-off?
4. From Figure 3, QVPO appears to converge faster. Can you provide an explanation or experiments to support this observation?
5. How does the computational complexity of QVPO compare with existing diffusion-based online RL methods in terms of training time and resource utilization?

**Limitations:**

The authors have addressed the limitations of the proposed method.

---

> ### Author Rebuttal · Authors · 2024-08-07
>
> We thank the reviewer for his/her careful reading and valuable suggestions. Below we will answer your concerns point-by-point.
>
> > **Q1**: The experiments are limited to D4RL datasets, which do not cover the full spectrum of possible RL environments. Can the proposed algorithm be adapted for tasks with discrete action spaces or sparse rewards, and if so, how?
>
> **A1**: Thanks for your question. We need to clarify two fundamental points:
>
> (1) **We never use D4RL datasets. QVPO is an online RL method instead of an offline one. Only offline RL requires datasets such as D4RL.** We perform our experiments in 5 mujoco online environments rather than D4RL datasets. These mujoco tasks are quite standard benchmarks in existing online RL methods. Besides, we also added the experiments on the recently released HumanoidBench environment in the attached pdf file. It is a very complex environment and QVPO still achieves the SOTA performance.
>
> (2) **QVPO is designed to handle complex continuous control tasks rather than for discrete action spaces or sparse rewards.** For discrete action spaces, we can directly sample in the discrete probability simplex output by the neural network and achieve multimodal policy. However, it is not trivial to achieve multimodal policy in continuous action distribution, since we cannot output the corresponding probability density function via neural networks. In that case, we consider utilizing diffusion model to enhance the multimodality and exploration ability of RL agent in continuous control tasks. Besides, QVPO does not conflict with most existing RL methods for sparse rewards like HER [R5]. Thus, we can incorporate HER into QVPO in environments with sparse rewards.
>
> > **Q2**:  While the paper includes ablation studies, they are not exhaustive. Additional ablation studies are needed to isolate the contributions of individual components, such as the Q-weighted variational loss and the entropy regularization term. It is unclear how much each component independently contributes to the overall performance improvements.
>
> **A2**: Thanks for raising the concern. We need to clarify that the main body of QVPO is the Q-weighted variational loss and QVPO must use this loss to train the diffusion policy. Thus, there is no way to do ablation study on the Q-weighted variational loss. Besides, the ablation study on the entropy regularization term is shown in Figure 4 of the original paper.
>
>
> > **Q3**: The proposed method for reducing diffusion policy variance via action selection aims to improve sample efficiency. However, this approach may sacrifice the exploration capabilities of the diffusion policy, thus limiting its multimodal potential. This trade-off is not discussed in the paper, and no ablation study is provided to evaluate the impact.
>
> **A3**: We did conduct such ablation study with the trade-off analysis in our paper. The action selection number $K_b$ in efficient diffusion policy is actually the trade-off between exploration and exploitation. As shown in **Figure 5** and mentioned in lines 320-330, the diffusion policy converges slowly when $K_b=1$ (i.e., without efficient diffusion policy). This means QVPO without efficient diffusion policy cannot exploit the information of Q value very well. Besides, we also add the experiment with $K_b=20$ to show that too high action selection number will lead to a limited exploration. In that case, setting $K_b=4$ can balance exploration and exploitation well.
>
> > **Q4**: How is the diffusion policy initialized, and what impact does this initialization have on the performance of the algorithm? If the policy is trained from scratch, how is the Q-function trained at the beginning stage?
>
> **A4**: Thanks for your question. The diffusion policy and Q-function are both trained from scratch without any special initilization trick.
>
> > **Q5**: The efficient diffusion policy method reduces policy variance but may limit exploration. How do you balance exploration and exploitation, and can you provide an ablation study to verify the effects of this trade-off?
>
> **A5**: The answer can be referred to in **A3**.
>
>
> > **Q6**: From Figure 3, QVPO appears to converge faster. Can you provide an explanation or experiments to support this observation?
>
> **A6**: Thanks for your question. Compared with DIPO and QSM, QVPO can explore broader action space and achieve multimodal training samples with the sampling-based update and entropy regularization term. We think that is why QVPO converges faster.
>
> > **Q7**: How does the computational complexity of QVPO compare with existing diffusion-based online RL methods in terms of training time and resource utilization?
>
> **A7**: The training and inference time of QVPO and other online RL methods are shown in **Global Rebuttal**. The GPU resource utilization rate of QVPO/DIPO is 55%/28% and memory consumption of QVPO/DIPO is 598MB/1354MB. In that case, QVPO can sufficiently make use of the parallel computing ability of GPU.
>
> [R5] Andrychowicz M, Wolski F, Ray A, et al. Hindsight experience replay[J]. Advances in neural information processing systems, 2017, 30.

---

> > ### Comment · Reviewer_Qp9e · 2024-08-10
> > **Response to rebuttal**
> >
> > Dear Authors,
> >
> > Thank you for your rebuttal. You mentioned that "QVPO can efficiently leverage GPU parallel computing, and the multiple-sampling and selecting procedure does not significantly impact time compared to gradient-based optimization like DIPO." Could you clarify the parallel settings used to evaluate the training and inference times for your method? Thank you.

---

> > > ### Author Response · Authors · 2024-08-12
> > >
> > > **Q8**: You mentioned that "QVPO can efficiently leverage GPU parallel computing, and the multiple-sampling and selecting procedure does not significantly impact time compared to gradient-based optimization like DIPO." Could you clarify the parallel settings used to evaluate the training and inference times for your method?
> > >
> > > **A8**: Thanks for your comments as well. As mentioned in lines 178-179, the difficulty of diffusion policy optimization is how to obtain the optimal action samples. QVPO finds the optimal action samples via multiple sampling and selection, which can be done parallelly. In contrast, the previous diffusion-based RL method DIPO finds the optimal action sample by performing multiple gradient updates on the actions in the replay buffer. However, the multiple gradient updates cannot be implemented in parallel. In that case, QVPO can speed up the training with the parallelism of GPU. The concrete parallel settings of QVPO in training and in inference are shown in Table 3 and lines 300-301 of the original paper respectively. The action selection number (i.e., the number of parallelly generated actions) is 64 in training and 32 in inference for each state.

---

> > > > ### Comment · Reviewer_Qp9e · 2024-08-13
> > > >
> > > > Thanks for your detailed rebuttal. The authors have addressed part of my concerns. I think the paper is slightly beyond the accept line. I will keep my score. Thank you.

---

### Official Review · Reviewer_Kypj · 2024-07-12

**Soundness:** 3
**Presentation:** 3
**Contribution:** 3
**Rating:** 6
**Confidence:** 5

**Summary:**

The paper uses a diffusion policy to address online RL. The method works by weighting the diffusion model VLO loss using advantages computed with a Q function. Additionally, they add an entropy maximization term to aid with exploration. The authors present results indicating strong performance on a variety of continuous control tasks as well as ablations verifying the necessity of several components of their method.

**Strengths:**

- The paper is one of the first to consider the application of diffusion policies to online RL and gets reasonably good results. This could be a significant paper with some additional experiments and touch ups.
 - The paper is the first to apply the weighted VLO objective to online RL and contributes a novel method for maximizing the entropy of a diffusion policy
 - Overall the discussion of the method and experiments is clear

**Weaknesses:**

There are several claims made in the paper that I don't believe are substantiated. Some important ones:
 - In the introduction and RW section you make claims about the DIPO and QSM baselines that don't seem to be proven anywhere
 - You take credit for several things that aren't yours. Your "efficient diffusion policy" trick is used extensively in other work and the weighted VLO loss is one of the methods presented in EDP (Kang et al.)

I think there need to be some changes to the experiments.
 - Most important: the comparison to other RL algorithms is unfair since you tune your algorithm per environment whereas other algorithms have a fixed set of hyperparameters. You need to either tune all relevant hyperparameters for all other algorithms for all environments or fix yours.
 - One of your key claims is that the diffusion policy is a good fit for RL because of its ability to model multimodal action distributions and yet you don't consider any environments where this is the case. Rather, you simply demonstrate improved sample efficiency on a handful of tasks that clearly can be solved using a MLP policy parameterization
 - You really should compare to QSM
 - The ablation studies are a bit incomplete in my opinion. It would be good to see results from all five environments. Also, it would be good to run an ablation where you omit the policy loss weighting, leading to an algorithm similar to IDQL, to verify that the policy weighting is important and that your policy isn't relying on the efficient diffusion policy step. I assume this ablation will go well for you based on the fact that the $K_b=1$ ablation still learns a policy but this will strengthen the paper.
 - You mention qcut, then say it doesn't work and never mention it again. Either show an experiment proving that or remove it from the paper
 - I believe that slow wall clock time, either for training or online rollouts, is potentially a big limitation of the method. Wall clock time is vaguely mentioned in the appendix but it would be good to see some data about this in the paper

**Questions:**

Random questions and comments
 - The second and third sentence of the abstract seem to contradict one another. How is it that diffusion policies can improve exploration (which is only relevant to online RL) if they have only really been applied to offline RL?
 - In the introduction you claim that DIPO has issues with limited exploration but I don't see any proof for that claim in your or their paper
 - In the introduction you claim that QSM has issues that prevent it from converging to optimal policies that your method does not have, but you never compare to QSM to demonstrate this win. I think QSM is an important comparison that really should be added to the paper since it is probably the closest to your method.
 - I don't understand the purpose of including qcut in the manuscript since you immediately propose a different approximation that works better and then never mention qcut again.
 - I'm very confused by Figure 2. How did you generate this? What does 'explorable area' mean?
 - Section 4.4 seems to imply that selecting an action by sampling several and then choosing the one that maximizes the Q value is a novel contribution, but this has been done on several occasions including by Diffusion QL, Efficient Diffusion Policy, Implicit Diffusion Q Learning, etc. This should be rewritten and these works should be cited. Also it is frequently referred to as "efficient diffusion policy" in the paper which is confusing since there is another paper (Kang et al.) that is called the same thing.
 - The policy improvement objective in this paper is actually very similar to one of the variants of EDP (Kang et al.), specifically shown in Equation 13 of https://arxiv.org/pdf/2305.20081. This paper still has reasonable novelty compared to that paper but that similarity should be better acknowledged.
 - Figure 4 is a nice result but it would be helpful to also see steps=100 & ent.
 - I'm curious about the wall clock training time of QVPO compared with baselines. Does everything take about 10 hours? I would expect QVPO to be quite a bit slower.
 - Is $\omega_\textrm{ent}$ difficult to tune?



Nit picks (not penalizing you for these but thought I should point them out):
 - The text in the plot legend is too small and the plots would be easier to read if the authors applied some smoothing
 - The transitions used at the beginning of the sentences in lines 163-170 (besides, firstly, notably, etc) are very distracting and the paragraph would sound better if you removed all of them.
 - There is an extra " on line 237 for some reason
 - Appendix B: hyperparameters should not be hyphenated
 - The legends of the plots look sloppy. Matplotlib allows the use of LaTeX and you should make use of that

**Limitations:**

The authors mention one limitation--the lack of adaptive entropy regularization. However, they miss other potential limitations, such as slower wall clock training time or the inability to output extended chunks of actions, as is common for diffusion policies applied to BC.

---

> ### Author Rebuttal · Authors · 2024-08-07
>
> > **Q1**: You make claims about DIPO and QSM that don't seem to be proven anywhere. You really should compare to QSM.
>
> **A1(DIPO)**: **We did compare QVPO with DIPO in Figure 4 (original paper)**. Besides, as mentioned in lines 110-114, DIPO creates a dedicated buffer for diffusion policy, updates the state-action pairs in this buffer via the gradient of Q network, and finally uses the state-action pair from this buffer to train the diffusion policy. In that case, the training samples of DIPO will be limited to the vicinity of previously explored actions, thus leading to limited exploration.
>
> **A1(QSM)**: The **code of QSM was not yet released when this paper was submitted**. So we did not compare QVPO with QSM before. We have **added the QSM results** of 5 Mujoco Benchmarks in Figure 2 of the attached pdf file, following the **recently released** official implementation. We can see the performance of QSM is much worse than QVPO and DIPO, and even worse than SAC and TD3 in the Walker and Humanoid environments, **which is also shown in Figure 3 of the QSM original paper**. As we mentioned in lines 43-46, QSM has a doubled approximation error from the alignment process. The trained Q value function has an approximation error. QSM utilizes the gradient of this approximated Q function to train the score model.
>
> > **Q2**: You take credit ... "efficient diffusion policy" trick is used extensively in other work and the weighted VLO loss is one of the methods presented in EDP (Kang et al.) ... This should be rewritten and these works should be cited. ... This paper still has reasonable novelty but that similarity should be better acknowledged.
> >
> **A2**: We will add more illustrations and cite these works properly in the final version. Note that these works are all offline RL. They used "efficient diffusion policy" to yield deterministic policy during inference, while our QVPO utilizes "efficient diffusion policy" in the online interaction with the environment to obtain **more efficient transition data**. Our motivation is different from these existing works.
>
> Besides, the weighted VLO loss of QVPO is different from EDP. In the weighted VLO loss, QVPO trains diffusion policy with the weighted sample from the current policy, while EDP trains the diffusion policy with the weighted sample from policy in the offline dataset (replay buffer).
>
> > **Q3-1**: The comparison to other RL algorithms is unfair. Need to either tune all relevant hyperparameters for other algorithms or fix yours.
>
> **A3-1**: Most hyperparameters in our experiments are the same, except $K_t$ in Hopper and $w_{ent}$ in HalfCheetah, as shown in Table 3. During the rebuttal, we **have fixed all hyperparameters for QVPO** and the **new results** are shown in the **attached pdf file for 5 Mujoco benchmarks**. QVPO shows similar performance and still outperforms all baselines with a clear margin.
>
> >**Q3-2**: **Is $\omega_{ent}$ difficult to tune?**
>
> **A3-2**: No, quite easy to set this value. Just make the scaling within 1e-3 to 1e-2.
>
> > **Q4**: Claim diffusion policy is a good fit for RL due to its ability to model multimodal action distributions, but don't conduct experiments on such environment.
>
> **A4**: Firstly, existing online RL methods almost all verify their performance on these robot locomotion tasks. In that case, we can compare QVPO with previous methods in a relatively fair manner with these standard environments.
>
> Second, the multimodality of policy is important because it can avoid the policy stuck at the local optimum. Moreover, our tested tasks such as Humanoid-v3 cannot be solved very well with a MLP policy.
>
> Third, to further test QVPO for more complex tasks, we applied QVPO to the recently released HumanoidBench environment. The benchmark is based on Unitree H1 humanoid robot with **151 state dimension** and **61 action dimension**. Please refer to **Global Response** for more details.
>
> > **Q5(1)**: The ablation studies are a bit incomplete in my opinion. It would be good to see results from all five environments.
>
> **A5(1)**: Due to the time and resource limit, we cannot complete these additional experiments. We will add them in our final version. Actually, the results for other 4 environments are similar but we did not record them.
>
> > **Q5(2)**: It would be good to run an ablation where you omit the policy loss weighting ... I assume this ablation will go well ... $K_b=1$ still learns a policy but this will strengthen the paper.
>
> **A5(2)**: The requested ablation study $K_b=1$ is shown in Figure 5 of the original paper. Offline diffusion RL methods like EDP train the diffusion policy with samples from offline datasets or replay buffer, while QVPO trains the diffusion policy with selected samples from the diffusion policy itself.
>
> > **Q6**: You mention qcut. Either show an experiment proving that or remove it.
>
> **A6**: In the beginning, qcut is mentioned to present the original motivation of QVPO. We will remove qcut part in the final version.
>
> > **Q7**: The training and inference time of QVPO.
>
> **A7**: Please refer to the **Global Response**.
>
> > **Q8**: The second and third sentence of the abstract seem to contradict one another.
>
> **A8**: Thanks and we will modify the two sentences.
>
> > **Q9**: I'm very confused by Figure 2. How did you generate this? What does 'explorable area' mean?
>
> **A9**: Figure 2 is exported by powerpoint to visualize the effect of diffusion entropy regularization term. The 'explorable area' means the area with a certain generation probability of diffusion policy.
>
> > **Q10**: Figure 4 is a nice result but it would be helpful to also see steps=100 & ent.
>
> **Q10**: The entropy term is introduced for the diffusion model with few denoising steps. QVPO with steps=100 with entropy term will not improve the result. We will add this result in the final version.
>
> > **Q11**: Suggestions to improve the presentation (some minor format issues) ...
>
> **A11**: We will modify them.

---

> > ### Comment · Reviewer_Kypj · 2024-08-07
> > **Most comments addressed but I'm worried about performance of all your baselines.**
> >
> > Thanks for your detailed responses. You addressed most of my concerns about the paper, but I was still not 100% convinced by your results. I thought it was odd that PPO does so poorly on Humanoid and HalfCheetah so I looked around for other papers running the same baselines on the same benchmark tasks. I found a paper titled "DSAC-T: Distributional Soft Actor-Critic with Three Refinements" from Duan et al. which also runs several of your baselines on your benchmark tasks. Notably, their implementation of SAC does at least as well as QVPO on all environments except Hopper, and actually does substantially better on Walker, Halfcheetah and Humanoid. This pattern hold across their implementation of several baselines you compare to. This leads me to question whether your method is truly stronger than these baselines or if the difference in performance that you observe is merely a matter of implementation details. In order to improve my score it would be helpful to understand why their results are so much stronger than yours.

---

> ### Author Response · Authors · 2024-08-08
> **Illustrations about the performance of our baselines ---- Reply to Reviewer Kypj**
>
> Thanks for your valuable suggestions as well. We are glad that our responses help you solve most of your concerns. Here are some further illustrations for your concerns.
>
> **Why results of DSAC-T are so much stronger than ours?**
>
> We want to clarify that the difference in performance between DSAC-T [R8] and QVPO merely **comes from different implementation settings**. As shown in Figure 2 of DSAC-T [R8], the x-axis is not the online interaction steps as in our paper (i.e., training epoch) but the training iteration. As shown in Table 3 of DSAC-T [R8], DSAC-T performs one training iteration with **20 online interaction steps** (**sample batch size in original paper**) for off-policy RL methods. In that case, to fairly compare QVPO with DSAC-T, you need to **multiply all the coordinates on the x-axis by 20**. With that operation, you will obtain results similar to ours on SAC. Actually, it is meaningful to use interaction steps (ours) rather than training iterations (DSAC-T) as the x-axis, since the former can show sample efficiency of different algorithms better in the online RL paradigm.
>
> We want to highlight that **our setting is the same as the most existing works**, including SAC [R8], DIPO [R9], QSM [R10], TD-MPC2 [R1], Dreamer-v3 [R2], etc. **Our results on all other baselines including SAC are also consistent with these works**. Please refer to Figure 1 of SAC [R8], Figure 6 of DIPO [R9], Figure 3 of QSM [R10], Figure 12 of TD-MPC2 [R1], etc.
>
>
> **Why PPO performs poorly on Humanoid and HalfCheetah?** PPO does perform poorly with limited interactions since it is an on-policy RL method. Many previous works such as DIPO [R7], SAC [R9], Dreamer-v3 [R2] also show similar results to ours. Please refer to Figure 6 in DIPO [R7], Figure 1 in SAC [R9], and Figure 14 in Dreamer-v3 [R9].
>
> [R8] Duan J, Wang W, Xiao L, et al. DSAC-T: Distributional soft actor-critic with three refinements[J]. arXiv preprint arXiv:2310.05858, 2023.
>
> [R9] Haarnoja T, Zhou A, Hartikainen K, et al. Soft actor-critic algorithms and applications[J]. arXiv preprint arXiv:1812.05905, 2018.
>
> [R10] Psenka M, Escontrela A, Abbeel P, et al. Learning a diffusion model policy from rewards via q-score matching[J]. arXiv preprint arXiv:2312.11752, 2023.

---

> > ### Comment · Reviewer_Kypj · 2024-08-08
> >
> > Thanks for the response. I incorrectly assumed that "iterations" in that paper was equal to the number of environment interactions. It appears you are correct that they are training on more data and your results do mostly match the DIPO results. Can you please confirm that in your paper one epoch equals one environment step? I couldn't find any details specifying this in the paper. I actually think replacing 'epoch' with something along the lines of the number of environment steps would get rid of a lot of confusion. At the very least you should make the relationship between epochs and training data more explicit in the paper.
> >
> > As I was looking into this I noticed that algorithm 1 is confusing. You loop over $t$ from 1 to $T$, but never specify what $T$ is. Is each iteration through that loop an epoch? Also, you use $t$ twice, as it simultaneously refers to the step in the iteration from 1 to $T$, (lines 1-3) but also refers to the index in the batch for the TD target (line 10).
> >
> > Regardless of the small issues I've raised in this response I'm feeling far more confident about the paper. Upon confirmation that one epoch means one environment step I'll increase my score to 6. The main reason I don't feel a higher score is justified is that I really need to see some experiments in more difficult environments for me to believe this paper will have "high impact", as is the criteria for a score higher than 6.

---

> > > ### Author Response · Authors · 2024-08-09
> > > **Clarification on the meaning of training epoch and the impact of QVPO -- Reply to Reviewer Kypj**
> > >
> > > **Q12: Does one training epoch equal one environment step in your paper?**:
> > >
> > > Thanks for your feedback. We confirm that one training epoch equals one environment step in our paper. We do not mention this since it is a common setting in the literature of online RL. However, to make our paper clearer, we will replace "training epoch" with "environment step" in the final version.
> > >
> > > **Q13: $t$ in Algorithm 1 looks confusing.**:
> > >
> > > Thank you for pointing out this ambiguity. In Algorithm 1, each iteration indicates a training epoch and $T$ means the total number of training epochs (i.e., 1e6 in our experiments on 5 mujoco environments). However, $t$ in line 10 does not mean the training epoch but represents different transitions in the replay buffer. We will use different notations for training epoch (environment step) in the final version.
> > >
> > > **Q14: I really need to see some experiments in more difficult environments for me to believe this paper will have "high impact"**:
> > >
> > > We need to emphasize that the HumanoidBench environment [R7] we added is difficult enough to present the "high impact" of QVPO. As mentioned in A4, This environment is based on Unitree H1 humanoid robot with **151 state dimension and 61 action dimension**. As shown in paper [R7], **model-free RL methods, such as PPO and SAC cannot even converge in this challenging environment**. However, as shown in **Figure 3 of our attached pdf file, QVPO (although as a model-free method) even surpasses the performance of advanced model-based RL methods (TD-MPC2, Dreamer-v3)**. Notably, the whole-body control of humanoid is truly very very hard! In that case, existing works on humanoid control such as [R11, R12] tend to separate the RL learning on lower body and upper body of the humanoid. However, QVPO can perform very well with whole-body humanoid control (61 action dimensions include the upper and lower body and two hands). Hence, we believe QVPO will be a revolutionary online RL method for complex continuous control tasks.
> > >
> > > [R7] Sferrazza C, Huang D M, Lin X, et al. Humanoidbench: Simulated humanoid benchmark for whole-body locomotion and manipulation[J]. arXiv preprint arXiv:2403.10506, 2024.
> > >
> > > [R11] He T, Luo Z, Xiao W, et al. Learning human-to-humanoid real-time whole-body teleoperation[J]. arXiv preprint arXiv:2403.04436, 2024.
> > >
> > > [R12] Cheng X, Ji Y, Chen J, et al. Expressive whole-body control for humanoid robots[J]. arXiv preprint arXiv:2402.16796, 2024.

---

### Official Review · Reviewer_nUs6 · 2024-07-13

**Soundness:** 3
**Presentation:** 3
**Contribution:** 3
**Rating:** 5
**Confidence:** 4

**Summary:**

The method proposes using Q-weight diffusion loss to train agents in online RL. Instead of approximating log probability, the paper uses advantage weight tuning to maintain the entropy of the actor.

**Strengths:**

The paper is well-organized, and the method is straightforward but effective. I found Theorem 1 interesting as it naturally builds a connection between diffusion models and the online RL objective. Experiments show the method is easy to converge and achieves higher rewards.

**Weaknesses:**

My main concern is that the paper has too much heuristic design regarding equivalent Q-weight transformation functions, the entropy regularization term, and Action Selection. Especially, Eq(9) and Eq(10) are called equivalent Q-weight transformation functions, but they are not mathematically equivalent to Eq(5).

**Questions:**

1. Equivalent Q-weight transformation functions are used to solve the negative Q value problem. Have the authors considered tuning rewards (for example, adding a fixed constant to all rewards to make all rewards positive) to guarantee Q values are all positive? It may be the most straightforward way to avoid the negative Q value issue.

2. There is a typo in line 208 where the right parenthesis should be "]".

3. For Eq(9), how do you estimate $\max_a Q(s,a)$ in practice?

4. Even though critic training is not the main point of the paper, I suggest adding a section in the Appendix to explicitly describe it for completeness.

5. In Figure 5, some $K_b,K_t$ combinations may lead to extreme training collapse. Did the authors experiment with higher values of $K_b,K_t$ while maintaining $K_t<K_b$?

**Limitations:**

My main concern is that the design of equivalent Q-weight transformation functions is heuristic and may need some theoretical intuition or backup.

---

> ### Author Rebuttal · Authors · 2024-08-07
>
> > **Q1**: My main concern is that the design of equivalent Q-weight transformation functions is heuristic and may need some theoretical intuition or backup.
>
> **A1**: Thank you for raising the concern. Here is the theoretical proof to show the convergence of QVPO with qadv weight transformation function. (We are afraid OpenReview cannot display the equation clearly. You can copy the following equations as well as texts into HackMD for a clear presentation.)
>
> Assume the new diffusion policy after one update can be approximately expressed as
> $$\pi_{new}(a\mid s) \approx (1-p_{data}(s)A_{\pi_{old}}(s, a^\star)\eta)\pi_{old}(a\mid s) + p_{data}(s) A_{\pi_{old}}(s, a^\star) \eta \frac{\mathbb{I}_{a\in \mathcal{N}(a^\star\mid s, \epsilon)}(a)}{S_{\mathcal{N}(a^\star\mid s, \epsilon)}}
> $$
> where $a^\star\mid s$ is the action that can maximize $Q_{\pi_{old}}(s, a)$ in the state $s$, $\mathcal{N}(a^\star \mid s, \epsilon)$ is the neighborhood of $a^\star\mid s$ with a small radius $\epsilon$, $S_{\mathcal{N}(a^\star\mid s, \epsilon)}$ is the area of this neighborhood, and $p_{data}(s)$ is the sampling distribution of the state. This assumption is straightforward: the training sample's generation probability in the diffusion model will be improved and the improved probability is proportional to its weight.
>
> Now consider the improvement of the RL objective:
> $$
> \begin{aligned}
>     \mathcal{J}(\pi_{new}) - \mathcal{J}(\pi_{old}) & = \mathbb{E}_{s\sim \rho_0}\left[V_{\pi_{new}}(s) - V_{\pi_{old}}(s)\right] \\
>     & = \mathbb{E}_{s\sim \rho_0}\left[\mathbb{E}_{a\sim \pi_{new}(a\mid s)}\left[Q_{\pi_{new}}(s, a)\right] - V_{\pi_{old}}(s)\right] \\
>     &=\mathbb{E}_{s\sim \rho_0}\left[\mathbb{E}_{a\sim \pi_{new}(a\mid s)}\left[Q_{\pi_{new}}(s, a)-Q_{\pi_{old}}(s, a)\right]\right] \\ &\quad \quad  + \mathbb{E}_{s\sim \rho_0}\left[\mathbb{E}_{a\sim \pi_{new}(a\mid s)}\left[ Q_{\pi_{old}}(s, a)\right] -  V_{\pi_{old}}(s)\right] \\
>     &=\mathbb{E}_{s\sim \rho_0}\left[\mathbb{E}_{a\sim \pi_{new}(a\mid s)}\left[Q_{\pi_{new}}(s, a)-Q_{\pi_{old}}(s, a)\right]\right] + \mathbb{E}_{s,a\sim \rho_0, \pi_{new}(a\mid s)}\left[A_{\pi_{old}}(s,a)\right] \\
> \end{aligned}
> $$
>
> The first term here can be further expanded according to the Bellman equation:
> $$
>     \mathbb{E}_{s\sim \rho_0}\left[\mathbb{E}_{a\sim \pi_{new}(a\mid s)}\left[Q_{\pi_{new}}(s, a)-Q_{\pi_{old}}(s, a)\right]\right] = \gamma \mathbb{E}_{s\sim d^1_{\pi_{new}}}\left[V_{\pi_{new}}(s)-V_{\pi_{old}}(s)\right]
> $$
> where $d^1_{\pi_{new}}$ denotes the probability distribution of state in time step $t=1$ with policy $\pi_{new}$. Repeating the above operation, we will obtain:
>
> $$
> \begin{aligned}
> \mathcal{J}(\pi_{new}) - \mathcal{J}(\pi_{old}) &= \sum_{t=0}^\infty \gamma^t\mathbb{E}_{s,a \sim d_{\pi_{new}}^t, \pi_{new}}\left[A_{\pi_{old}}(s,a)\right] \\
> &= \frac{1}{1-\gamma} \mathbb{E}_{s \sim d_{\pi_{new}}}\left[\mathbb{E}_{a\sim \pi_{new}(\cdot\mid s)}\left[A_{\pi_{old}}(s,a)\right]\right] \\
> &\approx \frac{1}{1-\gamma} \mathbb{E}_{s \sim d_{\pi_{new}}}\Bigg[(1-p_{data}(s)A_{\pi_{old}}(s,a^\star)\eta)\mathbb{E}_{a\sim \pi_{old}(\cdot\mid s)}\left[A_{\pi_{old}}(s,a)\right] \\& \quad \quad + p_{data}(s)A^2_{\pi_{old}}(s,a^\star) \eta \frac{\mathbb{I}_{a\in \mathcal{N}(a^\star\mid s, \epsilon)}(a)}{S_{\mathcal{N}(a^\star\mid s, \epsilon)}}\Bigg] \\
> & = \frac{1}{1-\gamma} \mathbb{E}_{s \sim d_{\pi_{new}}}\left[p_{data}(s)A^2_{\pi_{old}}(s, a^\star) \eta \frac{\mathbb{I}_{a\in \mathcal{N}(a^\star\mid s, \epsilon)}(a)}{S_{\mathcal{N}(a^\star\mid s, \epsilon)}}\right] \ge 0
> \end{aligned}
> $$
>
>
> > **Q2**: Have the authors considered tuning rewards (for example, adding a fixed constant to all rewards to make all rewards positive) to guarantee Q values are all positive?
>
> **A2**: Thank you for your question. Yes, we did conduct such a test by adding a fixed constant to all rewards at the very beginning. However, it does not work well after the policy has been improved to a certain extent. This is because the relative difference of Q value will be reduced with this operation. For example, if we have $Q(s,a_1)=1, Q(s,a_2)=2$, and $Q^{'}(s,a_1)=11, Q^{'}(s,a_2)=12$ after adding a fixed constant $10$, the raw relative difference is $\frac{2-1}{2}=0.5$ and the modified relative difference is $\frac{12-11}{12}=0.083$. Besides, in many real applications, how to set this fixed constant to ensure all $Q$ values positive is hard.
>
>
> > **Q3**: There is a typo in line 208 where the right parenthesis should be "]".
>
> **A3**: Thank you for pointing out this. We will correct this typo in the final version.
>
> > **Q4**: For Eq(9), how do you estimate $\max_a Q(s,a)$ in practice?
>
> **A4**: As mentioned in lines 219-220, we can obtain the approximate $\max_a Q(s,a)$ via enough samples from the diffusion. In practice, we choose around 64 samples.
>
> > **Q5**: Even though critic training is not the main point of the paper, I suggest adding a section in the Appendix to explicitly describe it for completeness.
>
> **A5**: Thank you for your valuable suggestion. The critic training part is standard and the same as TD3 and DiPO. We will add the critic training part to the appendix in the final version.
>
> > **Q6**: In Figure 5, some $K_b, K_t$ combinations may lead to extreme training collapse. Did the authors experiment with higher values of $K_b, K_t$ while maintaining $K_t<K_b$?
>
> **A6**: Thank you for your constructive suggestion. Firstly, we need to clarify that the fluctuation of the reward curve is a common phenomenon in RL and cannot be viewed as a training collapse, since it recovers quickly to the previous level. In Figure 5, we only plot the result of one run for each case without a window smoothing. That is why the reward curves look unstable.
>
> We have conducted experiments with higher $K_b$ and have presented the results in the attached PDF file. It can be found that a high value of $K_b$ will limit the exploration of QVPO and tend to be stuck at a local optimum.

---

> > ### Comment · Reviewer_nUs6 · 2024-08-10
> >
> > Thank you for the authors' detailed response. Most of my concerns have been addressed, but a few questions remain unresolved.
> >
> > 1. I believe the proof is valid. However, could you clarify what $\rho_0$ represents? Is it the discounted state distribution [1]? Additionally, could you explain the reasoning behind assuming the new policy in this manner? Do the authors have any intuition about this assumption?
> >
> > 2. I appreciate the explanation regarding reward tuning. One of my main questions following this is why not use $exp(Q)$ or $exp(A)$, which is a standard transformation when applying AWR-style methods, as referenced in [2,3]. This approach naturally guarantees that the weight is non-negative.
> >
> > [1] Sutton, R. S., & Barto, A. G. (1998). Introduction to reinforcement learning. mit press. Cambridge, MA.
> >
> > [2] Peng, X. B., Kumar, A., Zhang, G., & Levine, S. (2019). Advantage-weighted regression: Simple and scalable off-policy reinforcement learning. arXiv preprint arXiv:1910.00177.
> >
> > [3] Kostrikov, I., Nair, A., & Levine, S. (2021). Offline reinforcement learning with implicit q-learning. arXiv preprint arXiv:2110.06169.

---

> > > ### Author Response · Authors · 2024-08-12
> > >
> > > Thanks for your comments as well.
> > >
> > > **Q7**: I believe the proof is valid. However, could you clarify what $\rho_0$ represents? Is it the discounted state distribution [1]? Additionally, could you explain the reasoning behind assuming the new policy in this manner? Do the authors have any intuition about this assumption?
> > >
> > >  **A7**: As mentioned in lines 126-129 of our original paper, $\rho_0$ is the distribution of the initial state $s_0$ rather than the discounted state distribution. The discounted state distribution is represented by $d_\pi$.
> > >
> > > The assumption is straightforward. We will present a simpler formulation to make it easy to follow up. Generally, we can write the updated policy as $\pi_{new}(a\mid s) = (1-\beta(s))\pi_{old}(a\mid s) +
> > > \beta(s)\pi(a; a^\star\mid s)$ (similar update formula can be found in equation 4.1 of [R13]), where $\pi(a; a^\star\mid s)$ is a normalized probability distribution which is close to a Dirac distribution on $a^\star \mid s$ (i.e. we use $\frac{I_{a\in \mathcal{N}(a^\star\mid s, \epsilon)}(a)}{S_{\mathcal{N}(a^\star\mid s, \epsilon)}}$ to denote it). Considering weight $A_{\pi_{old}}(s, a^\star)$ in the objective, we just assume the degree of updating is proportional to the weight (i.e., $\beta(s)\propto A_{\pi_{old}}(s, a^\star)$). This assumption is reasonable that policy can achieve larger update steps in states with relatively larger weights.
> > >
> > > In that case, we will finally achieve $\pi_{new}(a\mid s) \approx (1-p_{data}(s)A_{\pi_{old}}(s, a^\star)\eta)\pi_{old}(a\mid s) + p_{data}(s) A_{\pi_{old}}(s, a^\star) \eta \pi(a; a^\star\mid s)$.
> > >
> > >
> > > **Q8**: I appreciate the explanation regarding reward tuning. One of my main questions following this is why not use $\exp(Q)$ or $\exp(A)$, which is a standard transformation when applying AWR-style methods, as referenced in [2,3]. This approach naturally guarantees that the weight is non-negative.
> > >
> > > **A8**: That is a good question. It can be observed that AWR [R3] converges slowly according to Figure 3 of AWR [R3]. One reason for the slow convergence is that $\exp(A)$ is too conservative for policy in online RL. For instance, if there exist 10 different actions for training, we have $A(s,a_1)=1$ and $A(s,a_i)=0.1, i=2,\cdots,10$, the weight of optimal action $a_1$ is $exp(1)\approx 2.7$ and the weight of sub-optimal actions is $\sum_{i=2}^{10}\exp(0.1)\approx 9.9$. In that case, the updated policy with $\exp$ weight function still has a high probability to output a sub-optimal action. In contrast, QVPO avoids this problem via the action selection procedure, which only uses the optimal action to train the diffusion policy. Another problem of $\exp(A)$ is the numerical instability. In practice, if the scale of reward is not properly set, the numerical instability will happen with $\exp(A)$ (e.g., $A(s, a)=10, exp(A(s, a))=22026$).
> > >
> > > [R13] Kakade S, Langford J. Approximately optimal approximate reinforcement learning[C]//Proceedings of the Nineteenth International Conference on Machine Learning. 2002: 267-274.

---

### Author Rebuttal · Authors · 2024-08-07

We thank the reviewers for their careful reading of considerate and meaningful suggestions to help us improve our paper. We sincerely appreciate that the reviewers find our work "straightforward but effective" (nUs6), "a significant work" (Kypj, Qp9e), "novel and innovative as the first to apply the weighted VLO loss to online RL" (Kypj, Qp9e), and contributes "a novel method for maximizing the entropy of a diffusion policy" (Kypj). We are further glad that the reviewers agree unanimously that our manuscript is "well-written and easy to read" (nUs6, Kypj) and confirm our contributions on both the "interesting and sound" theoretical analysis and "reasonably good, impressive, easy to converge" empirical results (Kypj, Qp9e, uSjP, nUs6) to support our algorithm.

In the following, we will try to address the concerns/questions of the reviewers and present a detailed item-by-item response to their comments.

**(1) Illustration of the attached pdf file**: The attached pdf file contains 5 figures.

Figure 1 replaces the schematic diagram in the original paper with an experiment on a continuous bandit toy example according to Reviewer uSjP. The contour lines indicate the reward function of continuous bandit, which is an arbitrarily selected function with 3 peaks. The concrete reward function is $Q(x)=\sum_{i=1}^3w_i\frac1{2\pi\sigma_i^2}\exp\left(-\frac1{2\sigma_i}(x-\mu_i)^T(x-\mu_i)\right)$, where $w_i=1.5$, $\sigma_i=0.1$, and $\mu_i=[-1.35,0.65]^T$, $[-0.65,1.35]^T$ and $[-1.61,1.61]^T$ respectively.

Figure 2 adds QSM as a new comparison baseline and shows rerun experimental results on QVPO with fixed hyperparameters. The legends and plots are all improved according to the suggestions of Reviewer Kypj.

Figure 3 adds experimental results on the recently released HumanoidBench [R4]. The benchmark is very complex based on Unitree H1 humanoid robot with **151 observation dimension** and **61 action dimension**, and **most existing model-free RL methods do not work totally in this environment.**  It verifies QVPO is competitive even compared with advanced model-based methods (e.g., TD-MPC2 [R1], Dreamer-v3 [R2]) in complex continuous control tasks that have high state and action dimensions.

Figure 4 is the comparison between QVPO with and without entropy term, which adds the experimental result of 5 diffusion steps with entropy term according to Reviewer uSjP. We believe 5 diffusion steps are not enough for diffusion policy in online RL.

Figure 5 is the comparison of QVPO with different action selection numbers for behavior policy $K_b$ and for target policy $K_t$, which adds the new case with $K_b=20, K_t=2$ according to Reviewer Qp9e. It implies that a too high $K_b$ in QVPO will result in a lack of exploration.

**(2) Comparison on training time and inference time**:
The training and inference time comparison is shown in the following tables. **Notably, since the official implementation of QSM is based on the jax and other algorithms are based on pytorch, it is not a fair comparison.** In practice, the same algorithm realized with jax is 6-10 times faster than that realized with pytorch. Besides, to fairly compare the diffusion-based RL methods in training and inference time, we set the same number of diffusion steps (T=20) for all of them (i.e., QVPO, QSM, and DIPO). The results imply that QVPO can sufficiently use the parallel computing ability of GPU and the multiple-sampling and selecting procedure does not take much time compared with gradient-based optimization like DIPO.

Moreover, we need to clarify that: **although QVPO is much slower than classical RL methods like SAC in inference, the inference time (6ms) still is acceptable**. To our knowledge, most existing real robots only require a 50-Hz control policy (i.e. output action per 20 ms). Besides, just like QSM, the inference time of QVPO can be further improved with jax framework if it is necessary. Hence, the inference time is not a bottleneck to applying QVPO to real applications.

It is worth noting that we only run one program at the same time here to avoid the effect of other running programs. That is why the training time of QVPO is less than what we mentioned in the original paper.

Due to the word limit, here we only show the comparison on Ant-v3. Results on other 4 environments are almost the same.

The training time comparison on Ant-v3 Benchmarks.
|  Method            |    QVPO   |     DIPO  |    TD3     |      SAC    |     PPO       |      SPO      | QSM (jax)  |
|:------------------:|:-------------:|:-------------:|:-------------:|:-------------:|:-------------:|:-------------:|:-------------:|
| **Training Time (h)**  |   6.8    |    10.5  |     0.5   |        2.5     |     0.3     |   0.3  |    1.0        |

The inference time comparison on Ant-v3 Benchmarks.
|  Method            |    QVPO   |     DIPO   |    TD3     |      SAC    |     PPO       |      SPO      | QSM (jax)  |
|:------------------:|:-------------:|:-------------:|:-------------:|:-------------:|:-------------:|:-------------:|:-------------:|
| **Inference Time (ms)**  |   6.2    |    5.7    |      0.2  |        0.3     |     0.2      |    0.3  |    0.9        |

[R1] Hansen N, Su H, Wang X. Td-mpc2: Scalable, robust world models for continuous control[J]. arXiv preprint arXiv:2310.16828, 2023.

[R2] Hafner D, Pasukonis J, Ba J, et al. Mastering diverse domains through world models[J]. arXiv preprint arXiv:2301.04104, 2023.

[R3] Peng X B, Kumar A, Zhang G, et al. Advantage-weighted regression: Simple and scalable off-policy reinforcement learning[J]. arXiv preprint arXiv:1910.00177, 2019.

[R4] Sferrazza C, Huang D M, Lin X, et al. Humanoidbench: Simulated humanoid benchmark for whole-body locomotion and manipulation[J]. arXiv preprint arXiv:2403.10506, 2024.

---

### Author Response · Authors · 2024-08-10
**Please let us know if any further questions! Thanks so lot!**

Dear Reviewers and AC,

Thank you for providing valuable comments. We have tried to address most if not all, concerns raised by the reviewers.

We observe that, except for Kypi, reviewers have no follow-up comments on our responses. Given the upcoming discussion stage deadline, please let us know if you have any further questions.

Besides, we would like to express our sincere gratitude to reviewer Kypi for recognizing our contribution and raising their score from 3 to 6.

Thank you and best regards.

Authors

---

### Decision · Program_Chairs · 2024-09-25

**Decision:**

Accept (poster)

**Comment:**

The reviewers appreciate the novelty of the work as one of the first results applying diffusion policies to online RL and obtaining reasonable results. However, they were all concerned about some limitations in the experiments and several claims made in the paper. Some of these concerns were addressed during the rebuttals and as a result reviewers raised their scores. However, it is still a borderline paper, slightly above the bar. I believe the novelty of the approach could be a good reason for acceptance but I strongly recommend that the authors take the questions/concerns raised by the reviewers into account in preparing the next revision of their work. Especially, it would be great if they can run some experiments in more difficult environments.